A model for correlation-based choreographic programming

Giallorenzo Saverio 1 2 saverio.giallorenzo@gmail.com
http://orcid.org/0000-0003-4666-901X Montesi Fabrizio 3
Gabbrielli Maurizio 2
1 INRIA , Sophia-Antipolis , France
2 Department of Computer Science and Engineering, University of Bologna , Bologna , Italy
3 Department of Mathematics and Computer Science, University of Southern Denmark , Odense , Denmark
Huisman Marieke
Electronic publication date: 2024 Dec 24
Publication date: 2024
Volume: 10
Electronic Location ID: e1907
Received 2023 Jun 28; Accepted 2024 May 24
Copyright: © 2024 Giallorenzo et al.
Copyright year: 2024
Copyright holder: Giallorenzo et al.
License: This is an open access article distributed under the terms of the Creative Commons Attribution License, which permits unrestricted use, distribution, reproduction and adaptation in any medium and for any purpose provided that it is properly attributed. For attribution, the original author(s), title, publication source (PeerJ Computer Science) and either DOI or URL of the article must be cited.
License URL: https://creativecommons.org/licenses/by/4.0/

Keywords: Concurrency, Distributed programming, Service-oriented computing, Choreographic programming

Funding: Independent Research Fund Denmark 0135-00219 Villum Fonden 29518 Innovation Fund Denmark 9142-00001B FREEDA I53D23003550006 Framework PRIN 2022 (MUR, Italy) French ANR project SmartCloud ANR-23-CE25-0012 This work was supported by the Independent Research Fund Denmark, grant no. 0135-00219, Villum Fonden, grant no. 29518, and Innovation Fund Denmark, grant no. 9142-00001B, by the research project FREEDA (CUP: I53D23003550006) funded by the framework PRIN 2022 (MUR, Italy), and by the French ANR project SmartCloud ANR-23-CE25-0012. There was no additional external funding received for this study. The funders had no role in study design, data collection and analysis, decision to publish, or preparation of the manuscript.

==============================
Choreographies provide a clear way to specify the intended communication behaviour of concurrent and distributed systems. Previous theoretical work investigated the translation of choreographies into (models of) programs based on message passing. However, existing theories still present a gap between how they model communications—using channel names à la CCS or π-calculus—and implementations—which use lower-level mechanisms for message routing. We start bridging this gap with a new formal framework called Applied Choreographies. In Applied Choreographies, developers write choreographies in a familiar syntax (from previous work) and reason about their behaviour through simple, abstract name-based communication semantics. The framework offers state-of-the-art features of choreographic models, e.g., modular programming supported via choreographic types. To provide its correctness guarantee, Applied Choreographies comes with a compilation procedure that transforms a choreography into a low-level, implementation-adherent calculus of Service-Oriented Computing (SOC). To manage the complexity of the compilation, we divide its formalisation and proof into three stages, respectively dealing with: (a) the translation of name-based communications into their SOC equivalents, namely, using correlation mechanisms based on message data; (b) the projection of the given choreography into a composition of partial, single-participant choreographies (towards their translation into SOC processes); (c) the translation of partial choreographies and the distribution of global, choreography-level state into local SOC processes. We provide behavioural correspondence results for each stage. Thus, given a choreography specification, we guarantee to synthesise its faithful service-oriented implementation.

Introduction

Background Concurrent, distributed software applications have become a crucial asset of our society. Messaging, governance, healthcare, and transportation are just some contexts recently revolutionised by distributed applications. The peculiarity of distributed applications is that their global behaviour, usually referred to as their protocol, emerges from the interaction of several programs, also called endpoints, that run in parallel and cooperate by means of message passing (Coulouris & Dollimore, 1988). Developers strive to correctly build each endpoint so that, when connected and run together, they faithfully enact the protocol that they should. If endpoints fail to follow their protocols, the distributed system can block or misbehave—e.g., due to deadlocks (Coffman, Elphick & Shoshani, 1971) or race conditions (Netzer & Miller, 1992).

Since the early days of distributed computing, designers and developers introduced and used tools to clearly specify the order of interactions among the endpoints of a system. Examples include the security protocol notation (Needham & Schroeder, 1978), Message Sequence Charts (International Telecommunication Union, 1996), and UML Sequence Diagrams (OMG, 2004). The common denominator of these tools is that they present a global description of the sequence of messages in the system. Recognising the usefulness of these global approaches, in the early 2000s a W3C working group defined a standard for describing interactions among Web Services. This resulted in the Web Services Choreography Description Language (WS-CDL) (W3C WS-CDL Working Group, 2004). A WS-CDL artefact is a choreography, which specifies the observable behaviour of all the endpoints involved in the system of interest, formalising from a global viewpoint the ordering and computation of the intended message exchanges.

Example 1. We illustrate choreographies with a representative example. We use the example to also introduce the syntax of choreographies used in the remainder of the article. The example describes a simple business scenario among a client process c, a seller service located at lS and a bank service located at lB. Locations ( l) are abstractions of network addresses, or URIs, which identify where services can be contacted to interact with them.

In Line 1, we find the start of a new instance of the protocol, called a session. In the example, the starter of the session is a process c, which plays the role of the client ( C, in square brackets). The process c sends a request to the respective locations of the seller ( lS) and the bank ( lB) services to create two new processes, respectively s, playing the seller ( S), and b, playing the bank ( B). Processes are distributed, i.e., they have separate, local states and run concurrently. Notice that the start command also has a parameter, k, which is the identifier of the (private) session where c, s, and b communicate over. Besides identifying the session (akin to cookies in Web browsers), here, we intend session identifiers as names that support the communication among the participants. We draw this interpretation from the line of work on Multiparty Session Types (MST) (Coppo et al., 2015), where, in a session, each process owns a statically-defined role, which identifies a message queue that the process uses to receive messages asynchronously for that session. Hence, e.g., looking at process c in the example, we assign to it the role C in session k. We interpret the establishment of a session as the point-wise connection of all the processes involved in it through the creation of message queues accessible under the session name. In the example, the process playing role C has one queue to receive messages from (the process playing) S and one from (the process playing) B; S and B have two dedicated queues as well—one to receive from C and one to receive from the other role. All these queues belong to k. The function of the session identifier is further clarified by its presence in all communication actions, which use the notation to declare in which session the communication takes place. The remainder of the communication action finds in the two placeholders on the left and right of the arrow, respectively, the sender’s and receiver’s information. Namely, we specify which participants interact, what expression we shall evaluate on the state of the sender to generate the outbound message, which operation1 of the receiver the interaction involves, and to what variable of the receiver we shall bind the content of the message.

Returning to the description of the example, in Line 2, the client invokes the operation buy of the seller, transmitting the name of a product it wishes to purchase. The seller stores that piece of data in its local variable x. In Line 3, the seller uses its internal function mk_order to prepare an order (e.g., compute the price of the product) and it asks the bank to open a payment transaction (on operation reqPay) for that order. In Line 4, the client sends its credit card ( cc) information to the bank on operation accPay. Then, in Line 5, the bank makes a local choice (also called internal choice) on whether it can transfer the credits from the client’s to the seller’s account (with the internal function confirm_pay, which takes the local variables cc and order as parameters). The bank then notifies the client and the seller of the outcome, by calling them on either operation ok or ko.

Example 1 illustrates a distributed application with three separate interacting programs in a clear, terse way. Indeed, the advantage of choreographies is their clarity; they succinctly and unambiguously specify the intended global behaviour of a distributed system made of communicating programs. For this reason, since the inception of WS-CDL, choreographies have been adopted also in other practical applications, like the Business Process Model and Notation by the Object Management Group (OMG, 2011) and Testable Architecture (JBoss Community, 2013). In general, choreographies come with the promise of enhancing correctness, since they equip programmers with precise specifications of what communications a system should enact. This promise motivated a fruitful line of research in the areas of process calculi and programming languages, which centers around the question “Can we use choreographies to prove that a concurrent, distributed program will execute exactly its intended interactions?”.

One way to try to answer positively to that question is, given the implementation of a set of endpoints, figuring out the protocol that emerges from their interaction and checking whether the former is compatible with the expected one. Unfortunately, ensuring that all endpoints play their respective parts correctly by looking at their possible interactions is difficult, due to the inherent non-determinism of programs running in parallel (O’Hearn, 2018). Specifically, inferring what protocol a set of given endpoints implement is computationally intractable. Indeed, algorithms for protocol inference have exponential complexity (Cruz-Filipe, Larsen & Montesi, 2017) even for simple systems with a fixed number of participants. However, checking the compatibility between the inferred and intended choreography is not the only explorable route and other two popular methodologies based on choreographies have emerged. The first is called Choreographic Programming (Montesi, 2013, 2023), and it interprets choreographies as programs. This kind of choreography has a syntax similar to the one shown in Example 1, and the idea is that they define both the internal computation performed by processes and the communications among them. Then, by equipping the choreographic language with a behaviour-preserving compiler, we can automatically synthesise (Carbone, Honda & Yoshida, 2012; Carbone & Montesi, 2013) correct-by-construction local endpoints that are guaranteed to faithfully follow the logic of the source choreography. In the second methodology, choreographies are used to describe protocols, which abstract away from the internal computation. The aim is to verify that each process, written manually (in contrast to being automatically synthesised, as in choreographic programming), implements correctly its role in the protocols that it participates in. MST (Hüttel et al., 2016) is a discipline representative of this methodology.

Both methodologies are based on the same general idea: for each endpoint described in a choreography, we can project a definition of its local behaviour using a procedure known as EndPoint Projection (EPP). In choreographic programming, EPP yields the local implementation of each endpoint. For MST, EPP produces a type for each endpoint, which one can use, e.g., to check that a process implementing that endpoint behaves according to its intended protocol. In both cases, the key technical result that one needs to prove is that the EPP always yields a set of endpoint terms (programs or types) that describe exactly the communications described in the source choreography. This is typically called the EndPoint Projection Theorem (or EPP Theorem, for short). The model of Compositional Choreographies (Montesi & Yoshida, 2013) unifies the two methodologies, to combine their advantages. In that model, programmers can describe parts of a system in choreographic programming and other parts as independent, local processes. The model uses MST to check that the execution of the independent processes with the projections of choreographic programs will behave correctly. The strong operational correspondence guaranteed by the EPP made the unification of the two approaches possible.

Motivation The main application area for choreographies so far is that of Service-Oriented Computing (SOC), as in web services (W3C WS-CDL Working Group, 2004) or microservices (Dragoni et al., 2017; Newman, 2015). Implementing communications in this setting is non-trivial, since services must be loosely coupled and one cannot assume the presence of any particular common middleware. However, in all previous definitions of EPP, both the choreography language and the target language abstract from how real-world frameworks support communications (Qiu et al., 2007; Lanese et al., 2008; Carbone, Honda & Yoshida, 2012; Carbone & Montesi, 2013; Carbone, Montesi & Schürmann, 2018; Cruz-Filipe & Montesi, 2020; Cruz-Filipe et al., 2022, 2023; Cruz-Filipe, Montesi & Peressotti, 2023; Montesi, 2023) and model message exchange through synchronisations on names (à la CCS/ π-calculus (Milner, 1980; Milner, Parrow & Walker, 1992a, 1992b)). Thus, implementations of choreographic frameworks (Chor Team, 2016; AIOCJ Team, 2016; Neykova & Yoshida, 2014; Choral Team, 2023) depart from their respective formalisations (Carbone & Montesi, 2013; Dalla Preda et al., 2015; Honda, Yoshida & Carbone, 2016; Giallorenzo et al., 2021) (a common aspect of implementing process calculi, cf. Carpineti, Laneve & Milazzo (2005), Hu, Yoshida & Honda (2008)). In particular, implementations realise the creation of new channels and message routing with additional data structures and message exchanges (Montesi, 2013; Dalla Preda et al., 2014) missing from their formalisations. The specific communication mechanism used in these implementations is message correlation. Correlation is the reference communication mechanism in SOC, where a message is relayed to a process/session/queue when a part of its content matches some data associated (i.e., correlated) with the process/session/queue. Mainstream technologies such as WS-BPEL (OASIS, 2007), Java/JMS, and C#/.NET support communication over message correlation. The gap between formalisations and implementations can compromise the correctness guarantees of choreographies. Thus, we ask: “Can we define a formal model of choreographies based on message correlation?”.

A satisfactory answer should find a way to preserve the correctness guarantees of the choreographic approach down to the level of how concrete communication mechanisms work. Defining such a model is challenging: we wish to retain the typical clarity of choreography languages, yet we need enough details to (formally) reason on how communications happen at the lower level. Ideally, the complexity of implementing communications should not leak into the choreographic programming model exposed to programmers, and should just be a “detail” that we can forget about with confidence. Building this confidence is the main aim of this article.

Contributions and outline

Concretely, we provide a positive answer to our research question by focussing on Compositional Choreographies (Montesi & Yoshida, 2013)—which we build upon to show that our approach applies to both the methodology of choreographic programming and that of MST—and by presenting a formal framework relying on a model for correlation-based choreographic programming.

We call our framework Applied Choreographies. In Applied Choreographies, developers abstract from the details of correlation-based communication, and rather write high-level choreographic programs using terse and informative choreographic syntax shown in Example 1. Then, a compilation process—consisting of a set of transformations into ever-more-involved intermediate representations and a tight series of correspondences (immaterial for the programmer)—generates a correspondent set of SOC endpoints that communicate using correlation and are guaranteed to faithfully implement the behaviour specified in the source choreography. To introduce the reader to the main components of Applied Choreographies, we represent them in Fig. 1, showing their position within the framework, the relevant properties that relate them, and where we present them in this article.

Figure 1 Schema of the components of the encompassing contribution of this article: a behaviour-preserving compiler from frontend choreographies to DCC distributed processes.

Frontend Choreographies The left-most artefact (①) in Fig. 1, are Frontend Choreographies programs (Frontend Choreographies). FC is the high-level, choreographic language Applied Choreographies provides to programmers. FC provides the elements developers are used to finding in a choreographic calculus; in particular, in FC, communications happen on name synchronisation, as in standard process calculi.

Backend Calculus and FC-to-BC transformation The first branch we find in Fig. 1 departing from FC (①) is that of items ② and ④. We start from ④, depicted as an intermediate artefact in Fig. 1, which is the second calculus that we present, called Backend Choreographies (BC) (Backend Choreographies). BC has the same syntax as FC but different semantics; instead of using abstract, name-based synchronisation, BC models and keeps track of the data structures needed to implement concrete SOC, correlation-based communications (Correlation-based Communication). While more involved than FC, BC is agnostic to the specific structure and technology that define the content of the data used for correlation. The other item, ②, is a transformation procedure (FC-to-BC) that generates the data structures needed to support the execution of a source FC program using message correlation (Encoding Frontend Choreographies to Backend Choreographies and Properties). Essentially, given an FC program, we obtain a BC one which is operationally correspondent to its source FC (Theorem 1).

EPP and Endpoint Frontend Choreographies The second branch departing from FC in Fig. 1 is that of items ③ and ⑤, which regard the EndPoint Projection (EPP) transformation. The latter allows us to transform an FC program that describes the behaviour of many participants in a set of artefacts written in a fragment of FC, called endpoint Frontend Choreographies (eFC), where an eFC program describes the behaviour of a single participant. More precisely, the EPP procedure is an endomorphism (Endpoint Projection (EPP)) that transforms a source FC program into a set of eFC programs, whose syntax is restricted to only partial actions (i.e., belonging to one of the two ends of a communication). The point of the EPP step is to act as a bridge between the global actions specified by FC and the local actions of endpoint processes. The result we prove in Theorem 2 is that the transformation performed by the EPP is guaranteed to generate a set of eFC programs which, run in parallel, behave like the source FC program. Hence, the EPP allows us to consider eFC as the syntax of the intermediate artefacts in the following steps of the compilation process.

Endpoint Backend Choreographies Since FC and BC share the same syntax, the next step in the Applied Choreographies pipeline ⑥ is to assemble the EndPoint Frontend Choreographies programs from item ⑤ with the BC data structures that support correlation-based communication ④ to obtain EndPoint Backend Choreographies, which we can proceed to compile into our target local implementations.

DCC and Compilation The third calculus is the target language for the compilation, called Dynamic Correlation Calculus (DCC) ⑧ (Dynamic Correlation Calculus). DCC is a process algebra of distributed executable code, based on a low-level formal model for SOC (Montesi & Carbone, 2011). DCC models both data distribution and how concrete correlation-based communications happen. Given its low-level scope, DCC does not capture all the abstraction of choreographies. The last item from Fig. 1 is the compiler ⑦ (Compiling Frontend Choreographies into DCC Processes), which takes in the eBC programs (at step ⑥) and it synthesises a behaviour-preserving implementation as a distributed system of DCC services.

Applied Choreographies Thanks to the step-by-step transformation correspondence results of steps ②, ③, and ⑦, we build our main contribution for Applied Choreographies, which is the definition of a behaviour-preserving compiler from Frontend Choreographies to DCC distributed services—the first correctness result of an end-to-end translation from standard choreographies to programs based on a real-world communication mechanism.

Our construction lets programmers use high-level programming primitives and semantics as found in previous work on choreographies—with state-of-the-art features like asynchronous communications (Carbone & Montesi, 2013) and modular development (Montesi & Yoshida, 2013)—while our compilation procedure tackles the heavy lifting of producing correct service-oriented implementations.

We conclude our proposal by discussing related and future work in “Related Work and Discussion” and report in the Supplemental Material auxiliary technical material and the proofs of our results.

This article integrates and extends material from Giallorenzo, Montesi & Gabbrielli (2018), where we present the main ideas behind the Applied Choreographies framework. Portions of this article were previously published as part of a preprint (Giallorenzo, Montesi & Gabbrielli, 2020) and the Ph.D. thesis of one of the authors (Giallorenzo, 2016). The extensions in this work include: (a) full formal definitions (syntax and semantics of all three calculi); (b) detailed examples for each main component of the work—the three calculi and the three stages of compilation—to illustrate their relevant characteristics and features; (c) full proofs of the formal properties guaranteed by the framework (in the Supplemental Material, to avoid breaking the flow of the reader with details of the technical development). Besides the previous points, this version contains an extended, revised, and refined presentation of all the contents presented in Giallorenzo, Montesi & Gabbrielli (2018).

Applied Choreographies: an overview. Before delving into the details of our contribution, we present, through a simple example, an overview of the three languages used in this work, their runtime, and their relationship as seen through the lenses of our compilation process. To structure the overview, we take the choreography from Example 1 and consider its first two instructions. Since the description would become quite involved if we described the evolution of the program from the first instruction (the “start”), in Table 1 we assume we performed the actions that start session k (which sets the state of processes and the needed queues up to support communication) and begin by describing the status of the FC program and its related translations in BC and DCC—for brevity, we focus on processes c and s and omit to represent process b. We let c send its message and comment on the changes in the state of these systems.

Table 1 Comparison between an FC source program and its BC encoding and DCC compilation.

	

In the first row called “Program” of Table 1, we find the FC program in the leftmost cell, in the central one the BC program—the same as FC—and in the rightmost one the corresponding DCC translation.

We recall that, in Applied Choreographies, FC is the only language (and semantics) exposed to the programmer, which uses it to implement the logic of a given distributed system. The other two languages, BC and DCC, are respectively a kind of intermediate representation to simplify the transition from high-level FC programs to lower-level, distributed services (in DCC) and the target language of the compilation process. Notice that below “Program” we find the “Deployment” row. Deployments introduce a remarkable difference between FC and pre-existing choreographic languages. To run an FC program we need to pair it with an FC deployment that describes the state of both its processes and session-based message queues. In the Table, we associate elements like process names and queues using pairs of the form (a,b). Specifically, we find that process c has the variable product pointing to the value ″book″, s has an “empty” state ( ∅), and both queues from C to S and S to C on session k—resp. referred by k[C⟩S] and k[S⟩C]—have no messages (the empty sequence ϵ).

Deployments allow us to transition from the name-based semantics of FC to the correlation-based one of BC without requiring the modification of the choreography. As visible from Table 1, the FC and BC programs are the same and what (considerably) changes from one model to the other is the shape of deployments. Indeed, since BC model a correlation-based communication semantics, we need to deal with more involved details: process locations (in the example, process c runs at the address clnt.com and process s at sllr.com), the data needed to support correlating messages with their intended queues, and the state of queues found at the different locations. Since the purpose of BC is to capture how communication works in SOC systems, which usually rely on XML- and JSON-formatted data (cf. “Correlation-based Communication”), we define the BC data model following a tree-like format. In Table 1, we see examples of this format for the state of processes c and s, where e.g., product_ is a leaf that points to the value ″book″. Let us focus on both branches k_ and their subtrees in the state of c and s. These structures represent the data used to communicate via correlation wrt a given session. For example, if we are process c and want to know how to find the queue where s is expecting to receive messages from us on session k, we can follow the path k.C.S_, we find ″X″ as the data structure (to keep this example simple, we use a string, but subtrees work too and the semantics of BC abstract away from this detail) that identifies/correlates with the queue. Since we assume (as in SOC) that the queue and the process that reads its messages are at the same location, we track, under the subtree k_, the location of each process; hence k.C.l_ both tracks the location of c and of all the queues it can receive messages on. The last element of BC deployments are message queues, which we identify from the combination of a location and some data. As expected, the queue at sllr.com that correlates with data ″X″—corresponding to the queue k[C⟩S] of the FC system—and the one at clnt.com that correlates with data ″Y″—corresponding to k[S⟩C] of FC—are empty ( ϵ).

Moving to DCC, the first striking difference we notice wrt FC and BC is that the status of the system (processes, queues) is not centralised into a single deployment, but it is distributed among the services that make up a network. Specifically, we find two services running in parallel ( |), resp. at the locations clnt.com and sllr.com. The services enclose two elements: a parallel composition of unnamed processes (for brevity, we omit inactive processes in the example and the corresponding composition with the parallel operator), defined by the combination of a behaviour ( Bc and Bs in Table 1) and a state ( tc and ts in Table 1). Since we already modelled a tree-shaped state for processes in BC, we adopt the same model for DCC, allowing us to take, unchanged, the state of BC processes for DCC ones. The second noticeable difference introduced by DCC is that actions are only “local”, e.g., the global FC/BC action k:c[C].product–⊳s[S].buy(x) is broken into a send ( Bc) and a reception ( Bs) actions in different processes. The syntax for communicating in DCC recalls the logic of message handling in BC. Indeed, the action in Bc sends a message on operation buy with the value of product_ to the queue in the service running at k.S.l_— sllr.com—and correlating with the data in k.C.S_— ″X″. In a complementary way, the action in Bs receives a message for operation buy, storing its payload under x_ from a queue within its enclosing service that correlates with k.C.S_—again, ″X″.

Closing our overview, we look at the bottom pair of rows in Table 1, after we let c (and its corresponding DCC process) send its message. For brevity, we report in these rows only the elements changed from the previous ones. At the level of choreographies (FC and BC), we reduced the program so that the next instruction we might execute is the reception by s of the received message. This kind of unfolding underlines how FC and BC model asynchronous communication, i.e., at runtime the global communication action breaks into the delivery of a message to a queue and its residual receive action in the program. Looking at the deployments of both FC and BC, we find the message sent by c in the corresponding queue for s. Similarly, in DCC we let the process at clnt.com send its message, so that Bc reduces to 0 (inaction). As expected, we find the message in the queue correlating with ″X″ at sllr.com.

Frontend choreographies

We present Frontend Choreographies (FC), the language model intended for programmers. Before giving the formal syntax of FC, we first describe the intuition behind its key components. Figure 2 displays the symbols that we are going to use, along with their names and domains.

Figure 2 Symbols and domains of the frontend choreographies calculus.

FC programs are choreographies, as in Example 1, denoted by C. A choreography describes the behaviour of some processes. Processes, denoted p,q∈P, are intended as usual: they are independent execution units running concurrently and equipped with local variables, denoted x∈Var. Processes communicate by exchanging messages. A message consists of two elements: (i) a payload, representing the data exchanged between two processes; and (ii) an operation, which is a label used by the receiver to determine what it should do with the message—in object-oriented programming, these labels are called method names (Pierce, 2002); in SOC, labels are typically called operations as in this article. Operations are denoted o∈O. Message exchanges happen through a session, denoted by k∈K, which acts as a communication channel. Sessions in FC are behaviourally typed (Hüttel et al., 2016). Intuitively, a session is an instantiation of a protocol, where each process is responsible for implementing the actions of a role defined in the protocol. We denote roles with A,B∈A. A process can create new processes and sessions at runtime by invoking service processes (services for short). Services are always available at fixed locations, denoted l∈L, meaning that they can be used multiple times (in process calculus terms, they act as replicated processes (Sangiorgi & Walker, 2001)).

FC supports modular development by allowing choreographies, say C and C′, to be composed in parallel, written C|C′. A parallel composition of choreographies is also a choreography, which can thus be used in further parallel compositions. Composing two choreographies in parallel allows the processes in the two choreographies to interact over shared location and session names.

We distinguish between two kinds of statements inside of a choreography: complete and partial actions. A complete action is internal to the system defined by the choreography, and thus does not have any external dependency. By contrast, a partial action defines the behaviour of some processes that need to interact with another choreography in order to be executed. Therefore, a choreography containing partial actions needs to be composed with other choreographies that provide compatible partial actions.

To exemplify the distinction between complete and partial actions, we consider the case of a single communication between two processes.

CompleteinteractionComposedpartialactionsk:c[C].product–⊳s[S].buy(x)k:c[C].product–⊳S.buy|k:C–⊳s[S].buy(x)

Above, on the left we have the communication statement as seen at Line 2 of Example 1. This is a complete action: it defines exactly all the processes that should interact ( c and s). On the right, we implement the same action as the parallel composition of two choreographies with partial actions: a send action by process c to role S over session k (left of the parallel) and a reception by process s from a role C (right of the parallel) over the same session k. More specifically, we read the send action (top of the parallel) as “process c sends a message as role C with payload product for operation buy to the process playing role S on session k”. We read the receive action (bottom of the parallel) as “process s receives a message for role S and operation buy over session k and stores the payload in variable x”. The compatible roles, session, and operation used in the two partial actions make them compliant. Thus, the choreography on the left is operationally equivalent to the one on the right. Observe that partial actions do not mention the name of the process on the other end—for example, the send action by process c does not specify that it wishes to communicate with process s precisely. This mechanism supports some information hiding: a partial action in a choreography can interact with partial actions in other choreographies independently of the process names used in the latter. Expressions and variables used by senders and receivers are also kept local to statements that define local actions.

Syntax of frontend choreographies

We present the formal syntax of FC, shown in Fig. 3. In the remainder, we use the symbol ∼ over an element as an ordered set of that kind of elements, e.g., p~ indicates an ordered set of processes p1,…,pn.

Figure 3 Frontend choreographies, syntax.

Complete Actions In term (start), process p creates a new session k together with processes q~ ( q~ is assumed non-empty). Process p, called active process, is already running, whereas each process q in l.q~, called service process, is dynamically created at the respective service location l. Each process is annotated with the role it plays in the new session k. Term (com) reads: on session k, process p sends to process q a message for its operation o; the message carries the evaluation of expression e on the local state of p, whilst x is the variable where q will store the content of the message. We leave the guest language for writing local expressions ( e) unspecified, and assume that it consists of terms for accessing local variables ( x) and implementing standard computations based on those (e.g., arithmetics).

Partial Actions A choreography can use partial actions to interact with other choreographies composed in parallel. Thus, Partial actions describe the behaviour of processes that wish to synchronise with “external” participants. Concretely, these external participants will be processes and/or services whose behaviour is defined in other choreographies composed in parallel. In (req), process p requests some external services, respectively located at l~, to create a new session k and some new external processes. Role annotations follow the same intuition as in term (start): in the new session k, p will play A and each new external process qi will play the respective role Bi. Term (acc) is the dual of (req) and defines a choreography module that provides the implementation of some service processes. In term (send), process p sends a message to an external process that plays B in session k. In term (recv), process q receives a message for one of the operations oi from an external process playing role A in session k, and then proceeds with the corresponding continuation. In the remainder, we omit curly brackets in (recv) when they have only one operation, i.e., k:A–⊳q[B].o(x);C is an abbreviation of k:A–⊳q[B].{o(x);C}.

Other Terms Term (seq) is sequential composition. In a conditional (cond), process p evaluates a condition e in its local state to choose between the continuations C1 and C2. Term (par) is standard parallel composition, which allows partial actions in two choreographies C1 and C2 to interact. Respectively, terms (def), (call), and (inact) model the definition of recursive procedures, procedure calls, and inaction. Some terms bind identifiers in continuations—the choreography that follows them in a sequential composition. In terms (start) and (acc), the session identifier k and the process identifiers q~ are bound (as they are freshly created). In terms (com) and (recv), the variables used by the receiver to store the message are bound ( x and all the xi, respectively). In term (req), the session identifier k is bound. Finally, in term (def), the procedure identifier X is bound. In the remainder, we omit 0 or irrelevant variables (e.g., in communications with empty messages). Terms (com), (send), and (recv) include role annotations only for clarity reasons; roles in such terms can be inferred, as shown in Montesi (2013).

Example 2. In Fig. 4, we extend (in blue) the behaviour of the seller of Example 1 to use an external module. In the updated code, the seller contacts an external service for the delivery of the product: the seller receives a request buy from the client, which now contains the wanted product along with the delivery address (Line 2). Next, the seller creates a new session k′ with an external delivery process (Line 3) and sends to the latter the shipping information of the product, e.g., the origin and destination addresses (Line 4). At Line 5, the seller receives the shipping costs, which it adds to the costs of the order at the bank (Line 6). At Lines 10 and 13, the seller notifies the delivery process if it shall ship the product or not. Let us call C1 the code above. We report in Fig. 5 the module C2 of a compliant delivery service for C1. We obtain a working system by composing the two choreographies in parallel: C1|C2.

Figure 4 Choreography C1, extension of example 1.

Figure 5 Choreography C2, compliant choreography to Fig. 4.

Semantics of frontend choreographies

We give an operational semantics for FC in terms of reductions of the form D,C→D′,C′, where D is a deployment. In FC, deployments keep track of the local states of processes (the values of their local variables) and the messages in transit in sessions, which we use to model asynchronous communications. While similar to other concepts, such as configurations and state, we name the runtime companions of choreographies “deployments” because they represent the environment where the software (the choreography) is deployed and executed, and we understand the term as more general than configurations/state—e.g., FC deployments contain both the state of processes and the configuration and state of queues. In the following, we formalise our notion of deployment for FC and we present its reduction semantics.

Frontend deployments

In the remainder, when indicating an FC program, we adopt as a convention its shortened form “Frontend choreography” (lowercase c) or simply “choreography” when the context clearly associates it to FC. We also use the shortened form “Frontend deployment” to indicate a Frontend Choreographies deployment.

To define Frontend deployments, we first define Q=K×A×A as the set of all queue identifiers. In FC, each pair of roles in a session has two asynchronous message queues that they can use to exchange messages (one per direction). We write k[A⟩B]∈Q to identify the queue from role A to role B in session k.

A Frontend deployment D is an overloaded partial function defined by cases as the sum of two partial functions, fs:P⇀Var⇀Val and fq:Q⇀Seq(O×Val) (their domains and co-domains are disjoint):

D(z)={fs(z)ifz∈Pfq(z)ifz∈Q

Function fs maps a process p to its state. A state is a partial function from variables x,y∈Var to values v∈Val. Function fq stores the queues used in sessions. Each queue is a sequence of messages m~=m1::…::mn|ϵ ( ϵ is the empty queue), where each message m=(o,v)∈O×Val contains the operation o for which the message is intended and the payload v. Deployments are a runtime concept: programmers do not need to define them, just as they normally do not explicitly give an initial state for their programs in other language models. Formally, we assume that choreographies without free session names start execution with a default deployment that contains empty process states. Let fp(C) return the set of free process names in C. Then, we formally define a default deployment as follows.

Definition 1 (Default Deployment). Let C be a choreography without free session names. Then, the default deployment D for C is defined as the function that maps all free process names in C to empty states (we write ∅ for the empty partial function from Var to Val): D=[p↦∅∣p∈fp(C)].

Intuitively, D is a default deployment for a choreography without free session names C if (i) D is defined for all and only the processes that appear free in C and (ii) the state of these processes is empty.

Frontend deployment transitions

In our semantics, choreographic actions have effects on the state of a system—deployments change during execution. At the same time, a deployment also determines which choreographic actions can be performed. For example, a communication from role A to role B over session k requires a queue k[A⟩B] to exist in the deployment of the system. We formalise the notion of which choreographic actions are allowed by a deployment and their effects using transitions of the form D,δ▸D′, read “the deployment D allows for the execution of δ and becomes D′ as result”. The following grammar defines δ actions.

δ::=startk:p[A]⊲–⊳l.q[B]~(sessionstart)|k:p[A].e–⊳B.o(sendinsession)|k:A–⊳q[B].o(x)(receiveinsession)

The rules defining D,δ▸D′ are given in Fig. 6.

Figure 6 Frontend choreographies, deployment transitions.

Rule ⌊D|Start⌉ creates a new session k between an existing process p and new processes q~ by updating the deployment with: a new (empty) state for each of the new processes q in q~ ( [q↦∅∣q∈q~]); and a new (empty) queue between each pair of distinct roles in the session ( [k[C⟩E]↦ϵ∣{C,E}⊆{A,B~}]).

Rule ⌊D|Send⌉ models the effect of a send action. In the first premise, we use the auxiliary function eval to evaluate the local expression e in the state of process p, obtaining the value v to use as message payload. Then, in the conclusion, we add a message (o,v)—where o is the operation used to label the message—to the tail of the queue k[A⟩B], i.e., the queue expected to contain messages sent by A to B in session k. We assume that function eval always terminates—in practice, this can be obtained by using timeouts.

Rule ⌊D|Recv⌉ models the effect of a reception. In the premise, we get the head of the message queue between the sender and receiver, i.e., (o,v), which we remove in the conclusion from the queue ( [k[A⟩B]↦m~]), updating the variable used to store the message in the state of the receiver ( [q↦D(q)[x↦v]]).

Reductions

We define the rules for reductions D,C→D′,C′ using deployment transitions. We call D,C a running choreography. The reduction → for FC is the smallest relation closed under the rules given in Fig. 7.

Figure 7 Frontend choreographies, semantics.

Rule ⌊C|Start⌉ creates a new session making sure that both the new session name k′ and processes r~ are fresh wrt D ( D#k′,r~). We use the fresh names in the continuation C (via standard substitution C[k′/k][r~/q~]).

Rule ⌊C|Send⌉ reduces a send action, if the deployment permits it: D,k: p[A].e–⊳B.o▸D′. Rule ⌊C|Recv⌉ reduces a message reception, if the deployment permits the reception of a message on one of the branches in the receive term ( j∈I). Recalling the corresponding rule ⌊D|Recv⌉, this can happen only if the deployment D has a message for operation oj in the queue k[A⟩B].

Rule ⌊C|Eq⌉ closes → under the congruences ≡C and ≃C. Structural congruence ≡C, reported in Fig. 8, is the smallest congruence supporting α-conversion, recursion unfolding, and commutativity and associativity of parallel composition. The swap relation ≃C, reported in Fig. 9, is the smallest congruence able to exchange the order of non-interfering concurrent actions. For example, provided pn returns the set of process names, Rule ⌊CS|EtaEta⌉ swaps two communications respectively enacted by completely disjoint processes. Rule ⌊C|Eq⌉ also enables the reduction of complete communications on (com) terms—see the last equivalence in Fig. 8, which unfolds a complete communication term into the two corresponding send and receive terms. Rule ⌊C|PStart⌉ starts a new session by synchronising a partial choreography that requests to start a session with other choreographies that can accept the request. The premise of the rule {l.B~}=⊎i⁡{li.Bi~}i, where ⊎ indicates the disjoint union of the list of located roles, requires that in the accepting choreographies the list of locations and their supported roles match the corresponding list of the request. The rest of the rule is similar to ⌊C|Start⌉. Conveniently, deployment transitions allow us to syntactically equate the effect of starting a session with either a complete start or the partial composition of partial actions. The choreographies accepting the request remain available for subsequent reuses.

Figure 8 Frontend choreographies, structural congruence ≡C.

Figure 9 Frontend choreographies, swap relation ≃C.

Rules ⌊C|Cond⌉, ⌊C|Ctx⌉, and ⌊C|Par⌉ respectively model guarded conditionals, recursion, and parallel composition in a standard way.

Example 3. The interplay between ≃C and rule ⌊C|Send⌉ yields an elegant formalisation of asynchronous behaviour for choreographies that, contrary to previous work (Carbone & Montesi, 2013), does not require a labelled transition system and ad-hoc rules. Consider Line 10 in Example 2, reported below.

C =def⁡ k:b[B]–⊳c[C].ok(); k:b[B]–⊳s[S].ok()

We can reduce C as follows (for brevity, we omit deployments):

C→k:B–⊳c[C].ok(); k:b[B]–⊳s[S].ok()by⌊C|Eq⌉ and ⌊C|Send⌉→k:B–⊳s[S].ok(); k:B–⊳c[C].ok()by ⌊C|Eq⌉ and ⌊C|Send⌉

In this case, process s may receive its message before process c, due to asynchronous message passing (the sending actions for process b are non-blocking).

Typing

Frontend Choreographies enjoy the standard type-safety guarantees of modern choreographic programming frameworks, in particular, the guarantee of deadlock freedom. Our typing checks the behaviour of sessions against protocols, given as MST. Interestingly, we retain the same syntax of traditional MST, yet we ensure that correct initial deployments do not corrupt at runtime due to inconsistencies among states and message queues. Since the main scope of this article regards the compilation process from Frontend Choreographies to DCC programs, in this section, we provide the main notions needed to understand the interactions the compilation process has with the typing environment. Section 1 of the Supplemental Material includes the full presentation (definitions, proofs, and examples) of the type system.

Briefly, we have a typing environment Γ that checks the conformance of a runtime choreography D,C. Notation-wise, we write Γ⊢D,C to indicate that D,C is well-typed under Γ. In particular, we make sure that the ensemble of a choreography C and its deployment D are coherent. For example, let us have C contain an already-started session k whereby a process p shall receive a message (with a certain type and label) from process q on session k. Our typing judgments look into D to verify the presence of the queue between p and q on k and that the messages therein (if any) correspond to the one that p is ready to receive from q. For reference, the terms found in the typing environment are: p.x : U, process p has a variable x holding a value of type U; X : Γ′, procedure X is typed by the environment Γ′; p : k[A], process p plays role A in session k; k[A]: T, role A in session k implements the type T; p@l, process p runs at location l; k[A⟩B]:T, types the messages in the queue where the process implementing role B in session k receives messages from role A; l~:G⟨A|B~|C~⟩, defines the type G of all sessions created by an active process playing role A which contacts the services at the locations l~. In the term, B~ are the roles pair-wise played by each service process while C~ are the roles implemented by the choreography that we are typing—we assume C~⊆B~, i.e., that C~ contain a subset of the roles in B~, ordered following the order in B~.

Besides type checking, Γ carries information useful for the compilation process, e.g., as presented in “Encoding Frontend Choreographies to Backend Choreographies and Properties”, we use the information carried by Γ to retrieve the information on the location, variables, and sessions of processes for building the deployment of Backend programs from a given FC deployment.

Backend choreographies

We now present Backend Choreographies (BC). The syntax of programs in BC is the same as that of FC. Also the two semantics are similar, except that FC communicate over named channels while BC formalises message exchange based on message correlation, as found in SOC (OASIS, 2007). Formally, thanks to the separation between choreographic programs and deployments presented in FC, we can let FC and BC share a large fragment of semantic rules, while the significant differences between the two semantics of message exchange—name-based for FC, correlation-based for BC—are isolated within their specific deployments and deployment transitions.

The structure and semantics of the deployments for Backend choreographies ID is one of our major contributions: it formalises, at the level of choreographies, how to implement sessions using the communication mechanism of message correlation typical of SOC systems. In the following section, we first informally introduce correlation-based message exchange, then we formalise data and queues in (the deployment of the) Backend Choreographies, and we formalise correlation-based message exchange in the semantics of deployment transitions in BC.

Correlation-based communication

Processes in SOC run within services and communicate asynchronously. To realise asynchronous communication, services provide an unbounded number of first-in-first-out message queues that processes interact with. The interaction happens from processes that associate a message insertion/retrieval action with a correlation key, which uniquely identifies the queue subject of the action. Concretely, a correlation key corresponds to a set of data that the service associates to a specific queue.

Processes retrieve messages from the queues of their enclosing service, as represented in (the right side of) Fig. 10 by process r1, which wants to consume a message received on queue Q1, associated to the correlation key k1. The request is satisfied by the service, which delivers message m1 to r1, also removing the interested message from the head of queue Q1. The complement of the action above is message insertion. Any process (within the queue-enclosing service and remote) can insert data into a queue by sending a message to the service owning the queue. That message must associate the payload with the correlation key that identifies the queue within the service. Concretely, when a service receives a message from the network, it inspects its content, looking for a valid correlation key, i.e., one that points to any of its queues. If a queue can be found, the message is enqueued in its tail. In Fig. 10, this is represented by data k1 marked by the attribute key in the message sent by process pn (of Service1) to Service2. At reception, Service2: (1) checks for the presence of the attribute key; (2) extracts the corresponding key k1; (3) finds the queue Q1, pointed by k1; (4) enqueues the received payload in Q1 as message mn.

Figure 10 Depiction of correlation-based message exchange in SOC.

As depicted in Fig. 10, messages in SOC contain correlation keys as either part of their payload or in some separate header. As in Montesi & Carbone (2011), also here we abstract away such details. To summarise, two processes can communicate over correlation-based messaging if: (i) the sender knows the (location of the) service where the addressee is running and (ii) the sender and the addressee know the key corresponding to a queue in the addressee’s service. After having presented the mechanism of correlation for message exchange, we can proceed to explain how we model SOC systems in BC.

Data and Process State. Data in SOC is structured following a tree-like format, e.g., XML (Bray et al., 1998) or JSON (Bray, 2017). In BC, we use trees to represent both the payload of messages and the state of running processes (as in, e.g., BPEL (OASIS, 2007) and Jolie (Montesi, Guidi & Zavattaro, 2014)).

Formally, we consider rooted trees t∈T, where T=Val∪L∪Set(Lab×T) and

t::=v|l|{x1―:t1,…,xn―:tn}

i.e., a tree (node) is either a value v, a location l, or a set of ordered pairs of edge labels x_,y_∈Lab and tree nodes. We assume tree nodes to be values or locations only in leaves. Given this definition of trees, we define BC variables as paths on trees (the latter, we remind, represent states of processes) as sequences of labels x,y∈Seq(Lab) such that x::=x_.x|ϵ, where ϵ is the empty sequence, which we often omit for brevity. When writing paths in their extended form, e.g., x_.y_.z_.ϵ, we often use the abbreviation x.y.z_.

In addition, we define two operators to handle trees: path application and deep copy. The path-application operator x(t) is used to access the sub-nodes pointed by path x in tree t. Intuitively, x(t) returns either the value, the location or the sub-tree pointed by path x in t. If x is not present in t, x(t) returns an empty set of ordered pairs label-tree. Formally,

x_.x(t)={x(x_.ϵ(t))ifx≠ϵt′ifx=ϵandt={x_:t′,…,xn_:tn}∅otherwise

The deep-copy operator t◃(x,t′) is a (total) replacement operator that returns the tree obtained by replacing in t the sub-tree rooted in x(t) with t′. If x is not present in t, t◃(x,t′) adds the smallest chain of empty nodes to t such that it stores t′ under path x. Formally,

t◃(x_.x,t′)={∅◃(x_.x,t′)ift∈Val∪L(t\{x_:x_(t)})∪{x_:t′}ift∉Val∪Landx=ϵ(t\{x_:x_(t)})∪{x_:x_(t)◃(x,t′)}otherwise

Backend deployments, transition rules, and BC semantics

On top of the convention of using the terms “Frontend choreography/deployment” to indicate a Frontend Choreographies program/deployment, in the remainder we adopt the same convention for “Backend choreographies” and “Backend deployments”. We use the term “choreography” alone, when the context makes it clear when we refer to Backend or Frontend choreographies. We define the notion of deployment for BC, denoted ID, which includes: (1) the locality of processes; (2) queues, pointed by a combination of a location and a correlation key; (3) the state of processes. Formally, ID is an overloaded partial function defined by cases as the sum of three partial functions gl:L⇀Set(P), gm:(L×T)⇀Seq(O×T), and gs:P⇀T. The domains and co-domains of the functions are disjoint, hence:

ID(z)={gl(z)ifz∈L,gm(z)ifz∈(L×T),gs(z)otherwise

Function gl maps a location to the set of processes running in the service at that location. Given a location l, we read ID(l)={p1,…,pn} as “the processes p1,…,pn are running at the location l”, assuming that each process p runs at most at one location. Function gm maps a pair location-tree to a message queue. This reflects message correlation as informally described above, where a queue resides in a service, i.e., at its location and is pointed by a correlation key. Given a pair l:t, we read ID(l:t)=m~ as “the queue m~ resides in a service at location l and is pointed by correlation key t”. The queue m~ is a sequence of messages m~::=m1::⋯::mn|ϵ and a message of the queue is m::=(o,t), where t is the payload of the message and o is the operation on which the message was received. Pairing operation labels with message payloads is typical of SOC implementations in general (as it happens, e.g., in SOAP messages (Mitra & Lafon, 2003)). Indeed, while not essential for the correct delivery of messages, operation labels are used by processes to program external choices (for instance, a process expecting to receive a message on either of two mutually-exclusive operations, e.g., to continue or exit a loop). The case applies also to BC, where we preserve the association between payload and operations (o,t)—similarly to FC (o,v). Function gs maps a process to its local state. Given a process p, the notation ID(p)=t means that p has local state t.

Backend Deployment Transitions In BC, we replace the deployment transitions of FC (commented in “Semantics of Frontend Choreographies”, rules in Fig. 7) with the rules for ID,δ▸ID′, reported in Fig. 11, explained below.

Figure 11 Backend choreographies, deployment transitions.

Rule ⌊ID|Start⌉ simply retrieves the location of process p (the one that requested the creation of session k) and uses rule ⌊ID|Sup⌉ to obtain the new deployment ID′ that supports interactions over session k. Namely, ID′ is an updated version of ID with: (i) the newly created processes for session k and (ii) the queues used by the new processes and p to communicate over session k. In addition, in ID′, (iii) the new processes and p contain in their states a structure, rooted in k_ and called session descriptor, which includes all the information (correlation keys and the locations of all involved processes) to support correlation-based communication in session k. Formally, this is done by rule ⌊ID|Sup⌉ where we ① retrieve the starter process, here called q1, which is the only process already present in ID. Then, given a tree t, we ensure it is a proper session descriptor for session k, i.e., that: ② t contains the location li of each process, represented by its role in the session Bi, under path Bi.l_; ③ t contains a correlation key tij for each ordered couple of roles Bi, Bj under path Bi.Bj_, such that ④ there is no queue in ID at location lj pointed by correlation key tij. Finally, we assemble the update of ID in four steps: ⑤ first, we obtain ID′ by adding in ID the processes q2,…,qn at their respective locations; ⑥ second, we obtain ID″ by adding to ID′ an empty queue ϵ for each pair lj:tij; ⑦ third, we obtain ID′′′ from ID′′ by storing in the state of (the starter) process q1 the session descriptor t under path k_; ⑧ and we update ID′′′ such that each new created process ( q2,…,qn) has in its state just the session descriptor t rooted under path k_. We deliberately define in ⌊ID|Sup⌉ the session descriptor t with a set of constraints on data, rather than with a procedure to obtain the data for correlation. In this way, our model is general enough to capture different methodologies for creating correlation keys (e.g., UUIDs or API keys). Rule ⌊ID|Send⌉ models the sending of a message. We comment on the premises. From left to right, the first gets the location l of the receiver B from the state of the sender p; the second retrieves the correlation key in the state of p (playing role A) to send messages to role B; the third evaluates the expression e of the sender p using its local state to get a value tm. The function eval evaluates expressions in a process state, traversing its paths and performing local computation. We highlight that, since in BC we preserve the syntax of Fronted Choreographies, we make two assumptions: that expressions (e.g., e in ⌊ID|Send⌉) are defined on Variables and that eval in BC automatically maps variables x, y, z into the respective paths x_.ϵ, y_.ϵ, and z_.ϵ, used to access the process states in ID. Finally, in the conclusion of the rule, we add the message (o,tm) in the queue pointed by l:tc that we found via correlation.

Rule ⌊ID|Recv⌉ models a reception. From left to right, the first premise finds the correlation key tc for the queue that q (playing role B) should use to receive from A in session k. The second premise retrieves the location l of q. The third accesses the queue pointed by l:tc and retrieves message (o,tm). The last premise updates ID to ID′ removing (o,tm) from the interested queue. Dually to rule ⌊ID|Send⌉, where eval maps variables into paths, in the conclusion of rule ⌊ID|Recv⌉ we map x, i.e., the intended variable that should store the payload tm in the state of q, into path x_.ϵ.

Encoding frontend choreographies to backend choreographies and properties

Now that we have presented Backend Choreographies, we can proceed with defining a compilation procedure from high-level FC programs to low-level services. Here, we tackle the transition from FC programs to their intermediate representation toward SOC systems as Backend Choreographies. Specifically, we translate FC programs that use the abstract mechanism of communication over names, into BC programs that use the concrete mechanism of correlation-based communication. We prove our translation correct, i.e., that our encoding guarantees an operational correspondence between the semantics of a Frontend choreography and its Backend encoding. Formally, since choreographies in BC have the same syntax as FC ones, we can translate FC runtime terms D,C to BC runtime terms by encoding the FC deployment D to an appropriate Backend deployment. Below, we define the encoding in the form of an algorithm for clarity and compactness. Notably, BC deployments contain more information wrt FC deployments. We extract this data from Γ, the typing environment of D,C.

Definition 2 (Encoding FC in BC). Let Γ⊢D,C and ⟪D⟫Γ be defined by the algorithm in Fig. 12. Then, the Backend encoding of D,C is defined as ⟪D⟫Γ,C.

Figure 12 Encoding algorithm from frontend to backend deployments.

What the algorithm ⟪D⟫Γ does is: 1. include in ID all (located) processes present in D (and typed in Γ); 2. translate the state (i.e., the association Variable- Value) of each process in D to its correspondent tree-shaped state in ID; 3. for each ongoing session in D, set the proper correlation keys and queues in ID and, for each queue, import and translate its related messages.

More precisely, in the algorithm defined in Fig. 12 at Line 1, we create a new Backend deployment ID and assign to it the totally undefined function ( ∅); ID is an empty Backend deployment. Then, following Lines 2–13, we make the following updates on ID: Lines 2–4, for each located process p@l in Γ, we update the locations of ID to contain p at location l (Line 4) and we include process p in ID, associating to it an empty state, i.e., the empty tree ∅ (Line 4); Lines 5–6, for each variable x (typed in Γ) of a process p, we update the state of process p in ID to include the association of x to its value in the state D(p). As done in rules ⌊ID|Send⌉ and ⌊ID|Recv⌉, we map FC variables x∈Var into BC paths x_∈Seq(Lab); Lines 7–13, follow the same principles to support correlation-based exchanges as formalised in rule ⌊ID|Sup⌉; for each couple of processes p,q, respectively playing distinct roles A and B in a session k, with q located at l. Lines 8 , we obtain a fresh correlation key t with auxiliary function fresh. The latter takes deployment ID and location l as input and returns a correlation key which is fresh among the keys associated to location l in ID. Formally t is such that l:t∉dom(ID); Lines 9, we associate correlation key t with location l in ID and make it point to the corresponding queue of messages from role A to role B in D (accessed with triple k[A⟩B]). Note that we can directly copy message queues from D into ID. Indeed, while message queues in D and ID are respectively of type Seq(O×Val) and Seq(O×T), by definition T subsumes Val; Lines 10–11, we include in the state of processes p (Line 10) and q (Line 11) correlation key t, storing it under path k.A.B_; Lines 12–13, we include in the state of processes p (Line 12) and q (Line 13) the location of role B under path k.B.l_.

The encoding from FC to BC guarantees a strong operational correspondence.

Theorem 1 (Operational Correspondence (FC ↔ BC)). Let Γ⊢D,C. Then:

1. (Completeness) D,C→D′,C′ implies ⟪D⟫Γ,C→⟪D′⟫Γ′,C′ for some Γ′ s.t. Γ′⊢D′,C′;

2. (Soundness) ⟪D⟫Γ,C→ID,C′ implies D,C→D′,C′ and ID=⟪D′⟫Γ′ for some Γ′ s.t. Γ′⊢D′,C′.

Note that, since we use Γ to carry the information needed to synthesise a BC deployment from an FC one, such that Γ⊢D,C, we are restricting the set of FC programs that we encode to BC to only the well-typed ones. For example, in well-typed Frontend choreographies, no process can be present in more than one choreography composed in parallel and, thus, there is inter-process but no intra-process parallelism. In general, we would not need to impose one such restriction for supporting the encoding of FC deployments to BC ones. However, in the later steps of our compilation pipeline, we restrict the class of compilable programs to well-typed ones. Thus, we prefer to avoid introducing a dedicated environment for this step—which would further involve our development—and we rather adopt the minimal solution of supporting the encoding via the typing environment.

We report in Section 3.3 of the Supplemental Material the proof of Theorem 1. Intuitively, we can prove (Completeness) by induction on the derivation of D,C. The main observation is that the encoded system ⟪D⟫Γ,C mimics D,C by applying the same semantic rules on C and the corresponding deployment transitions (e.g., respectively defined by rules ⌊D|Send⌉ and ⌊ID|Send⌉). Let ID′ be the Backend environment obtained from the reduction ⟪D⟫Γ,C→ID,C′ on rule ⌊C|Start⌉. Since the encoding algorithm ⟪D⟫Γ (cf. Fig. 12) and the rule ⌊ID|Sup⌉ (on which rule ⌊ID|Start⌉ relies) implement the same principles, we know that k.A.B_(ID) and k.A.B_(⟪D′⟫Γ′) will be the same, except possibly for (i) the location of processes and (ii) trees of correlation keys corresponding to the same paths. Concretely, item (i) derives from the fact that Γ and Γ′ can disagree on the location of the same process p, and item (ii) is caused by the random generation of correlation keys, for which, considering a correlation key rooted in k.A.B_ of a process p, the trees obtained from k.A.B_(ID(p)) and k.A.B_(⟪D′⟫Γ′(p)) may differ. However, these discrepancies do not constitute a problem, since both locations and correlation keys are used consistently in their respective deployments, which are thus interchangeable. This observation allows us to surmise that, without loss of generality, we can consider the case that Γ and Γ′ agree on the location of the services and that the random generation of the correlation keys coincides, making the equation hold.

The same holds for (Soundness), which we can prove by induction on the derivation of ⟪D⟫Γ,C.

Dynamic correlation calculus

We introduce the Dynamic Correlation Calculus (DCC), the target language of our compilation.

DCC is an extension of a previous proposal called Correlation Calculus (Montesi & Carbone, 2011), which is a process calculus that formalises service-oriented, correlation-based communications. Indeed, while we started this work considering CC as the target language of our compilation, we found it limited for our purposes: in CC each process receives from only one message queue, while we need processes to be able to select receptions from multiple queues (as in our Backend deployments). Hence, we defined DCC as an extension of CC with the support for the dynamic creation and selection of queues in processes.

We deem DCC a choice that fits the practical motivations of this work thanks to its closeness to the implementation languages/frameworks listed below, which informs how we can apply our theoretical results to future implementations. First, CC formalises the semantics of message exchange of Jolie, a service-oriented programming language (Montesi, Guidi & Zavattaro, 2014). Thus CC specifications are directly translatable into Jolie executable programs. This is not the case for our DCC code, as Jolie lacks the primitives to let processes create and select queues. Fortunately, CC and DCC are similar enough so that supporting the extended features in Jolie would entail minimal changes, i.e., the inclusion of the syntactic primitives for queue creation and selection2 and the implementation of the associated semantics—a direct extension of the one-process-one-queue semantics of the current implementation. Second, the service-oriented language BPEL (OASIS, 2007) lets processes create and receive from multiple queues, making DCC a useful reference for BPEL-based implementations. Third, besides service-oriented languages, DCC abstracts real-world message-exchange models where processes can interact with multiple message queues—as in some actor models (Agha, 1985) that associate one actor with many queues/mailboxes (Haller & Odersky, 2007) and in some popular message-exchange middlewares (Vinoski, 2006; Videla & Williams, 2012), which are suitable alternatives to the implementation targets above.

Syntax We now introduce the syntax of DCC, which we report in Fig. 13 and which comprises two layers: Services, ranged over by S, and Processes, ranged over by P. In the syntax of services, term (srv) is a service, located at l, with a Start Behaviour B and running processes P (both described later on) and a queue map M. The queue map is a partial function M:T⇀Seq(O×T) that, similarly to function gm in Backend deployments, associates a correlation key t to a message queue. We model messages like in BC where a message is a couple (o,t), o being the operation on which the message has been received, and t the payload of the message. Services are composed in parallel in term (net).

Figure 13 Dynamic correlation calculus, syntax.

Concerning behaviour, DCC distinguishes between start behaviour and process behaviour. Process behaviour defines the general behaviour of processes in DCC, as described later. Start behaviour uses the term !(x) to indicate the availability of a service to generate new local processes on request. At runtime, the reception of a dedicated message triggers the start behaviour B of a service and the creation of a new process. The new process has (process) behaviour B, which is defined in B after the !(x) term, and an empty state. The content of the request message is stored in the state of the newly created process, under the bound path x. As in BC, also in DCC paths are used to access process states.

Finally, processes (prc) in DCC consist of a behaviour B and a state t and can be composed in parallel (par). Process states t are trees. In Behaviour, operations ( o), procedures ( X), paths ( x), and expressions ( e, evaluated at runtime on the state of the enclosing process) are all the same as defined for Backend Choreographies (Correlation-based Communication). Terms (input) and (output) model communications. In (input), the process stores under x a message from the head of the queue correlating with e and received on operation o. The term (output) sends a message on operation o. The three expressions in the term define: e1, the location of the service where the addressee is running; e2, the content of the message; e3, the key that correlates with the receiving queue of the addressee. Term (choice) is an (input)-choice: when one of the inputs can receive a message from the queue correlating with e on operation oi, it discards all other inputs and executes the continuation Bi. Term (reqst) is the dual of (acpt) and asks the service located at e1 to spawn a new process, passing to it the message in e2. Term (newque) models the creation of a new queue that correlates with a unique correlation key (in the service hosting the running process). The correlation key is stored under path x in the state of the process, for later access. The remaining terms are standard.

Semantics In Fig. 14, we report the rules defining the semantics of DCC, a relation → closed under a (standard) structural congruence ≡D that supports commutativity and associativity of parallel composition. We comment on the rules. Rules ⌊DCC|Assign⌉, ⌊DCC|Ctx⌉, and ⌊DCC|Cond⌉ are standard for, respectively, assignments, procedure definition, and condition evaluation. Rule ⌊DCC|PEq⌉ uses equivalence ≡D on DCC processes to describe parallel execution and recursion. The rules of ≡D are reported in the lower part of Fig. 14. Rule ⌊DCC|Newque⌉ adds to M an empty queue ( ϵ) correlating with a randomly generated key tc. The key is stored under path x of the process that requested the creation of the queue. As in rule ⌊ID|Sup⌉ of BC (see “Correlation-based Communication”), we do not impose a structure for correlation keys, yet we require that they are distinct within their service. Rule ⌊DCC|Recv⌉ models message reception. Since both (input) and (choice) define receptions of messages, we consider both cases in the rule. Indeed, the first premise of the rule captures the presence of either an (input)—with shape oj(x)frome—or a (choice)—with shape ∑i∈I[oi(xi)frome]{Bi}. In both cases, we obtain the correlation key of the receiving queue from the evaluation of expression e against the state of the receiving process ( t). We inspect queue map M and check if it has a message in its head received on operation oj. If this condition holds, the rule removes the message from the queue and stores the payload ( tm) under path xj in the state of the process.

Figure 14 Dynamic correlation calculus, semantics.

Regarding message delivery, in DCC, there are two output actions: (i) (output) used by a process to communicate with another one and (ii) (reqst) used by a process to require the creation of a new process in a service. Since in DCC communications can happen within the same service or between two services, we describe two sets of rules, either for internal and inter-service message delivery. We start from the easier case of internal delivery, defined by rules ⌊DCC|InSend⌉ and ⌊DCC|InStart⌉. In rule ⌊DCC|InSend⌉ a process B⋅t sends a message into a queue of its hosting service. This requirement is embodied by the second premise of the rule, where the location l, corresponding to the evaluation of expression e1 against the state of the sender process, is the same as its hosting service. As expected, correlation key tc must point to an actual queue of the service. This is checked by the last premise, which requires tc to be in the domain of queue map M. In the conclusion of the rule, we update the content of the queue pointed to by tc including message (o,tm) in its tail. In rule ⌊DCC|InStart⌉, a service accepts the request to create a new process from one of its local processes. In the conclusion of the rule, we find the newly created process Q. The behaviour of the new process corresponds to the one associated with the (acpt) term of the service ( B′). The state of the new process is empty ( ∅) except for the inclusion of the payload of the request, stored under path x and obtained from the evaluation of e2 against t. The rules ⌊DCC|Send⌉ and ⌊DCC|Start⌉ define message delivery between two services. The two rules are similar to their respective internal cases, except for requiring the location defined by the sender (i.e., the one obtained from the evaluation of expression e1 against the state t of the sender process) to match that of the receiving service. The last two rules in Fig. 14 are ⌊DCC|SPar⌉ and ⌊DCC|SEq⌉ and define the (parallel) execution of networks of services.

Compiling frontend choreographies into dcc processes

We now present our main result: the correct compilation of high-level FC into low-level DCC networks of services (and processes). Recalling the schema presented in Fig. 1, given an FC program D, C and its typing environment Γ, the stages involved in the compilation from FC to DCC programs include:

FC-to-BC (② in Fig. 1) the encoding, defined in “Encoding Frontend Choreographies to Backend Choreographies and Properties”, of the Frontend deployment D to a correspoding Backend deployment ID=⟪D⟫Γ;

EPP (③ in Fig. 1) the projection of the choreography C into a parallel composition of partial choreographies (i.e., where actions concern only one participant), each defining the behaviour of a single active or service process in C. This stage, presented in “Endpoint Projection (EPP)”, is called Endpoint Projection;

Compilation (⑦ in Fig. 1) the compilation of the composition of the results of the previous stages—essentially, an endpoint Backend choreography—into a network of corresponding DCC services and their located processes. We present the compilation in “From Backend Endpoint Choreographies to DCC (Compilation)”.

The three-stage division simplifies the definition of the compilation process and its related correctness checks. In particular, they ease the extraction of the behaviour of a single process (EPP) from the source FC and of its state (FC-to-BC) from the source Frontend deployment. In the remainder of this section, we detail the projection stage (EPP), we define how we pair the outputs of FC-to-BC and EPP and the properties of that pairing, and we present the Compilation stage and the related properties.

Endpoint projection (EPP)

Given a choreography3 C, its Endpoint Projection (EPP), denoted ⟦C⟧, returns an operationally-equivalent composition of Endpoint choreographies. Intuitively, an Endpoint choreography is a choreography that does not contain complete actions—i.e., terms (start) and (com)—and that describes the behaviour of a single process. We also recall that a choreography can contain two kinds of processes: active processes which are already running, and service processes which accept requests to create new active processes at their respective associated location l. As detailed later on, our EPP procedure projects Endpoint choreographies onto all processes, both active and service ones.

Our definition of EPP is an adaptation of the one presented in Montesi & Yoshida (2013) and it is divided into two components: (1) a process projection that derives the Endpoint choreography of a single process p from a given choreography C, denoted ⟦C⟧p; (2) the actual EPP of a given choreography C, which results in the parallel composition of: (2a) the process projections of all active processes in C; (2b) the process projections of all service processes in C, with the exception that we merge into the same Endpoint choreography all projections of service processes that accept requests at the same location.

We first present the process projection and then the actual Endpoint Projection.

Process Projection We define process projection starting from formalising Endpoint choreographies.

Definition 3 (Endpoint Choreographies). Given a (Frontend/Backend) choreography C. If either: (a) C=acck:l.q[B];C′, and q is the only free process name in C′; (b) C has only one free process name. Then C is an Endpoint choreography.

The process projection of a subject process p in a choreography C, returns the Endpoint choreography obtained following the rules defined in Fig. 15. Process projection follows the structure of the source choreography. We briefly comment the rules in Fig. 15, from top to bottom. We start with the complete actions (start) and (com) which, if the subject process takes part in them, are projected onto proper partial terms. When projecting a (start) action, if the subject process is the active process p, we project a (req). Otherwise, if the subject process is one of the service processes in q~, we project an (always-available) (acc). Similarly, when projecting a (com) action, if the subject process is the sender or the receiver in the interaction, we respectively project a (send) or a (recv). Partial actions (acc), (req), and (send) are projected verbatim, except for (acc) terms, which define the availability of only the subject process. When projecting a (rec) term, we project both the body of the procedure ( C′) and the choreography C. This is safe even if r does not take part in the body of X; indeed, in that case, the projection of C′ is just an (inact) term. As a consequence, we can safely project (call) terms verbatim. The projections of conditionals and receptions are peculiar: we project a conditional verbatim if the subject process evaluates the condition; for all other processes, we merge their behaviour with the merging (partial commutative) operator ⊔, defined by the rules reported in Figure 8 of the Supplemental Material. We define C⊔C′ only for Endpoint choreographies, returning a choreography isomorphic to C and C′ up to receptions including all receptions with distinct operations. We use ⊔ also in the projection of (recv), where we require the merging of the behaviour of all processes not receiving the message. The projection of two choreographies in parallel is the parallel composition of their projections and (inact) is projected verbatim.

Figure 15 Frontend choreographies, process projection.

Note the definition of the rule of process projection for (rec) terms. Indeed, applying a naïve rule like

⟦defX=C′inC⟧r=defX=⟦C′⟧rin⟦C⟧r

in the EPP would yield more than one procedure with the same identifier, which could prevent the obtained projection from being typable as, according to the typing rules defined in “Typing”, we cannot have in Γ two definition typings on the same identifier. To tackle the issue, the rule for (rec) terms in Fig. 15 guarantees the coherent definition and usage of process-unique identifiers through renaming. The renaming is safe as, by assumption, we consider well-sorted choreographies where definitions always precede recursive calls. We conclude this paragraph with the formal definition of process projection.

Definition 4 (Process Projection). ⟦C⟧r is a partial homomorphism from (Frontend/Backend) choreographies to Endpoint Choreographies, inductively defined by the rules in Fig. 15.

Endpoint Projection We proceed to define our Endpoint Projection. In the definition below, we use the grouping operator ⌊C⌋l, which returns the set of all service processes accepting requests at location l. We report in Figure 9 of the Supplemental Material the rules that inductively define ⌊C⌋l.

Definition 5 (Endpoint Projection). Let C be a (Frontend/Backend) choreography. The endpoint projection of C, denoted by ⟦C⟧, is defined as:

⟦C⟧=∏p ∈ fp(C)⟦C⟧p⏟(i) | ∏l(⨆p ∈ ⌊C⌋l⟦C⟧p)⏟(ii)

Definition 5 states that the EPP of a choreography is the parallel composition of two kinds of Endpoint choreographies: (i) Endpoint choreographies that are the process projection of active processes p∈fp(C) and (ii) Endpoint choreographies that are the merge ( ⊔) of the process projections of all service processes available at the same location l, i.e., p∈⌊C⌋l.

Example 4. As an example of Endpoint Projection, let C be the choreography at Lines 5–9 of Example 1 (for convenience, we report the mentioned snippet of code grayed-out in the lower part of Fig. 16). The EPP of C, ⟦C⟧, is the parallel composition of the process projections of processes c, s, and b, i.e., respectively ⟦C⟧c, ⟦C⟧s, and ⟦C⟧b. As per Definition 5, ⟦C⟧=⟦C⟧c | ⟦C⟧s | ⟦C⟧b.

Figure 16 Endpoint projection of lines 5–9 of example 1.

We report in the top half of Fig. 16 the projections ⟦C⟧c, ⟦C⟧s, and ⟦C⟧b. The example is useful to illustrate that the projection of the conditional is homomorphic on the process ( b) that evaluates it. The projection of a (com) term results into a partial (send) for the sender—as in the two branches of the conditional in ⟦C⟧b—and a partial (recv) for the receiver—as in ⟦C⟧c and ⟦C⟧s. Note that the EPP merges branching behaviour: in ⟦C⟧c and ⟦C⟧s the two complete communications are merged into a partial reception on either operation ok or ko.

Properties

We conclude this section by presenting the guarantees provided by the Endpoint Projection wrt to the source Frontend choreography, as formalised in Theorem 2. Before presenting Theorem 2, we introduce the notion of pruning (as defined in Carbone, Honda & Yoshida (2012)), where ≺ specifies an asymmetric relation between two choreographies C and C′, written C≺C′, in which C prunes some unused accepts and receptions of C′. To give a formal definition to our pruning relation, we present the two concepts of subtyping of typing environments and minimal typing system. Below we just give the intuition on both concepts, which are formalised in the Supplemental Material. First, given two typing environments Γ and Γ′, Γ is a subtype of Γ′, written Γ≺Γ′, if Γ is identical to Γ′ up to (i) some local and global types that are more constrained in Γ than in Γ′ and (ii) some service typings present in Γ′ and not present in Γ. We report the formal definition of Γ≺Γ′ in Definition 10 of the Supplemental Material. Second, the minimal typing system Γ⊢minC uses the minimal global and local types to type sessions and services in C. We report in Section 3.4.1 of the Supplemental Material the formal definition of minimal typing.

We can finally formalise the pruning relation.

Definition 6 (Pruning). Let Γ⊢minC and Γ′⊢minC′ , if Γ≺Γ′ then C prunes C′ under Γ, written Γ⊢minC≺C′, or C≺C′ for short.

The shortened form C≺C′ is similar to Carbone, Honda & Yoshida (2012), where, as in this article, it does not lose precision since it is always possible to reconstruct appropriate typings. The pruning of C′ by C means that C omits unused inputs and service processes present in C′. The ≺ relation is thus a strong bisimulation since C≺C′ means that the two choreographies have precisely the same observable behaviour, except for the receive actions at pruned receptions and unused available service processes. In our definition, we use the term projectable to indicate that, given a choreography C, we can obtain its projection ⟦C⟧. Formally

Definition 7 (Projectable Choreography). Let C be a choreography, we call C projectable if there is a choreography C such that C′=⟦C⟧.

We can now write the statement of our EPP Theorem.

Theorem 2 (EPP Theorem). Let D,C be a well-typed FC program with C projectable. Then,

1. (Well-typedness) D,⟦C⟧ is well-typed.

2. (Completeness) D,C→D′,C′ implies D,⟦C⟧→D′,C′′ and ⟦C′⟧≺C′′.

3. (Soundness) D,⟦C⟧→D′,C′′ implies D,C→D′,C′ and ⟦C′⟧≺C′′.

When projecting choreographies in Theorem 2, we assume that these are well-sorted. We also note that the requirement of projectability and well-typedness of C in Theorem 2 implies that parallel compositions in C (if any) happen outside (cond) and (recv) terms since the projection of nested parallels are undefined—the merge operator ⊔ (cf. Figure 8 of the Supplemental Material) does not define how to reconcile different branches with parallel compositions—and that no process can be present in more than one choreography composed in parallel. Moreover, we make one assumption that trades the simplicity of technical development off of the expressiveness of FC programs that are supported by Theorem 2. Namely, we assume, when present, that (acc) terms are at the top level, i.e., not preceded by other terms in sequential compositions. Morally, requiring top-level (acc) terms corresponds to having service processes always available. Technically, accommodating for non-top-level (acc) terms would make our treatment and proofs more complex—e.g., one would need to extend the swap relation (cf. Fig. 9) so that we can match the behaviour of top-level (acc) terms generated by the projection (and composed in parallel) with swap actions to hoist the corresponding term in the original choreography. As mentioned, we prefer the simplicity of treatment (and proofs) over the coverage of cases that FC can capture—that, we underline, are not typical of the SOC context, such as services that become available after some preceding actions.

We report in Sections 3.4–3.7 of the Supplemental Material the proof of Theorem 2. In short, we prove Theorem 2 by presenting the minimal typing system for FC (proving its existence) and the projection of typing environments, so that, given the minimal typing environment of a choreography C we can build the minimal typing environment for the EPP of C. We prove the property of well-typedness of Theorem 2 by proving the stronger result of typing preservation (Theorem 5 of the Supplemental Material) between C and its EPP under the minimal typing system. To prove the remaining properties of Theorem 2, we introduce lemmas on the invariance of the EPP wrt the swap relation (Fig. 9) and structural congruence (Fig. 8), and on the distributivity of EPP over parallel composition. We re-state the last two items of Theorem 2 (Completeness and Soundness) in terms of an annotated semantics of FC, which allows us to precisely characterise the operational correspondence between the source and projected choreographies.

From backend endpoint choreographies to DCC (Compilation)

This is the last stage of our compilation process, where, given a parallel composition of Backend Endpoint choreographies, we obtain a network of DCC services that faithfully follows the semantics of the source choreography. Given a Backend deployment ID, a parallel composition of endpoint choreographies C, and a typing environment Γ, we write ID,CΓ to indicate the compilation of ID,C under Γ into DCC. To formalise ID,CΓ, we use the auxiliary functions: C|l which acts as a filter on C to get the endpoint choreography in C of the service process accepting requests at location l (e.g., C|l=acck:l.p[A];C′′); C|p which acts as a filter on C to return the endpoint choreography in C participated only by process p; CΓ, given a single endpoint choreography C and a typing environment Γ, compiles C to DCC, using the rules in Fig. 17; l∈Γ, a predicate satisfied if, according to Γ, location l contains or can spawn processes; ID|l returns the partial function of type T⇀Seq(O×T) that corresponds to the projection of function gm in ID with location l fixed. Formally, for each t such that ID(l:t)=m~, ID|l(t)=m~.

Figure 17 Compiler from endpoint choreographies to DCC.

Definition 8 (Compilation). Let ID be a Backend deployment, C a parallel composition of endpoint choreographies, and given the typing environment Γ

ID,CΓ=∏l∈Γ⟨C|lΓ,∏p∈ID(l)C|pΓ⋅ID(p),ID|l⟩l

Intuitively, for each service ⟨B, P, M⟩l in the compiled network: (i) the start behaviour B is the compilation of the endpoint choreography in C accepting the creation of processes at location l; (ii) P is the parallel composition of the compilation of all active processes located at l, equipped with their respective states according to ID; (iii) M is the set of queues in ID corresponding to location l. We comment on the rules in Fig. 17, where the notation ⊙ is the sequence of behaviour ⊙i∈[1, n](Bi)=B1;…;Bn.4

Requests Function start defines the compilation of (req) terms, which generates the code to create the queues and a part of the session descriptor for the starter (this is similar to what rule ⌊ID|Sup⌉ does in Backend deployment transitions, cf. “Correlation-based Communication”). Given a session identifier k, the located role of the starter ( lA.A), and the other located roles in the session ( lB.B~), start returns the DCC code that: (1) [ s1] includes in the session descriptor all the locations of the processes involved in the session; (2) [ s2] for each role, except for the starter: (2a) creates the key and the correlated queue that the current role will use in the session to communicate with the starter; (2b) requests the creation of the service process that will play the current role in the session; (2c) waits on the reserved operation sync to receive the correlation data for the session defined by the newly created process; (3) [ s3] sends to the newly created processes the complete session descriptor obtained after the reception (in the sync step) of all correlation keys.

Accepts (acc) terms define the start behaviour of a spawned process at a location. Given a session identifier k, the role B of the service process, and the service typing G⟨A|C~|D~⟩ of the location, function accept compiles the code that: ( a1) accepts the request to spawn a process, ( a2) creates its queues and keys, updates the session descriptor received from the starter, and sends it back to the latter ( a3). Finally with ( a4) the new process waits to start the session.

Other terms A (send) term compiles to a DCC (output) term. Notably, the compiled code contains the same elements used by the semantics of BC to implement correlation, i.e., the location of the receiver ( k.B.l_) and the key that correlates with its queue ( k.A.B_). Similarly, (recv) compiles to (choice), which defines the path ( k.A.B_) of the key correlating with the receiving queue.

Example 5. As an example of compilation, we compile the first two lines of the choreography C in Example 1, considering a deployment ID and a typing environment Γ such that Γ⊢ID,C.

 ID,[[C]]Γ=⟨0,Pc⟩lC | ⟨BS,0⟩lS| ⟨BB,0⟩lBwhere

Pc={k.S.l―=lS;k.B.l―=lB;ν⟩k.S.C―;?@k.S.l―(k―);sync(k―)fromk.S.C―;ν⟩k.B.C―;?@k.B.l―(k―);sync(k―)fromk.B.C―;start@k.S.l―(k―)tok.C.S―;start@k.B.l―(k―)tok.C.B―;/∗end of start−request∗/buy@k.S.l―(product)tok.C.S―;…and

BS={!(k―);ν⟩k.C.S―;ν⟩k.B.S―;sync@k.C.l―(k―)tok.S.C―;start(k―)fromk.C.S―;/∗ end of accept ∗/buy(x)fromk.C.S―;…

We omit BB, which is similar to BS.

Properties of applied choreographies

We close the section by presenting our main result, i.e., a compiler from FC to DCC networks and its properties. Theorem 3 defines our result, for which, given a well-typed, projectable Frontend choreography, we can obtain its correct implementation as a DCC network. Such a result is obtained by merging the properties of the stages FC-to-BC (Encoding Frontend Choreographies to Backend Choreographies and Properties), EPP (Properties), with our Compilation procedure (From Backend Endpoint Choreographies to DCC (Compilation)). In the definition, we use the translation steps defined earlier. Namely, we encode FC deployments to BC deployments, written ⟪D⟫Γ, as per Definition 2, we project choreographies into endpoint choreographies with the endpoint-projection operator ⟦C⟧ from Definition 5, and we translate the endpoint Backend choreographies thus obtained via the compiler (⟨⟨D⟩⟩Γ,[[C]]Γ) from Definition 8.

Theorem 3 (Applied Choreographies). Let D,C be a Frontend choreography where C is projectable and Γ⊢D,C for some Γ. Then:

1. (Completeness) D,C→D′,C′ implies

⟨⟨D⟩⟩Γ,[[C]]Γ→+⟨⟨D′⟩⟩Γ′,C′′Γ′and[[C′]]≺C′′andforsomeΓ′,Γ′⊢D′,C′

2. (Soundness) ⟨⟨D⟩⟩Γ,[[C]]Γ→∗S implies

D,C→∗D′,C′andS→∗⟨⟨D′⟩⟩Γ′,C′′Γ′and[[C′]]≺C′′andforsomeΓ′,Γ′⊢D′,C′

We report in “Conclusion” of the Supplemental Material the proof of Theorem 3. The salient points of the proof include lemmas that prove that renaming free variables with fresh names in processes (and, by extension, in services) preservers bisimilarity and the usage of the annotated semantics of FC (introduced for the proof of Theorem 2) for the more precise characterisation of the operational correspondence.

The last result we provide is a corollary of the properties of our typing discipline, which guarantees that well-typed Frontend choreographies are deadlock-free (cf. Theorem 3 of the Supplemental Material), and Theorem 3, which allows us to state that deadlock-freedom is preserved from well-typed choreographies to their final translation in DCC. The definition uses the predicate co(Γ), which holds if and only if (i) each session and the related typings follow their corresponding global type G, (ii) all needed services to start new sessions are present, and (iii) all the roles in every open session are correctly implemented by some processes. We say that a network S in DCC is deadlock-free if it is either a composition of services with terminated running processes or it can reduce.

Corollary 1. Γ⊢D,C and co(Γ) imply that D,[[C]] Γ is deadlock-free.

Related work and discussion

Applications of choreographic programming include cyber-physical systems (López & Heussen, 2017; López, Nielson & Nielson, 2016), security protocols (Bruni et al., 2021; Lluch-Lafuente, Nielson & Nielson, 2015), and distributed agreement (Jongmans & van den Bos, 2022).

One of the main lines of work on the paradigm regards its growth into a general approach for concurrent and distributed programming, focusing in particular on the synthesis/verification of sets of local programs that comply with choreographies—first explored using automata or process calculi abstractions (Alur, Etessami & Yannakakis, 2003; Qiu et al., 2007; Basu, Bultan & Ouederni, 2012; Honda, Yoshida & Carbone, 2016; Autili, Inverardi & Tivoli, 2018; Autili et al., 2020). The earliest implementations of choreographic programming languages consist of Chor (Carbone & Montesi, 2013) and AIOCJ (Dalla Preda et al., 2017). These generate executable Jolie code but their models, based on process calculi (resp. by Carbone & Montesi (2013) and Dalla Preda et al. (2015)), do not capture the low-level, correlation-based semantics of the target language, leaving a gaping space between the formalisation and the implementation. Choral (Choral Team, 2023) is a more recent interpretation of choreographic programming married to an object-oriented approach. The language abstracts away the media and formats used to support communication, which are parametric wrt to the source program and compiled system. Choral also lacks a specific theoretical model and existing work only formalised its main constructs following a functional approach (Cruz-Filipe et al., 2022) or introduced minimal models to compare it with other, existing paradigms for concurrent, distributed systems (Giallorenzo et al., 2021). Other implementations, such as Pirouette (Hirsch & Garg, 2022) and HasChor (Shen, Kashiwa & Kuper, 2023), conjugate choreographic programming in a functional setting. HasChor is a library for functional choreographic programming in Haskell that, like Choral, lacks a dedicated formal model. Pirouette is a higher-order functional choreographic programming language formalised in Coq whose compilation target is a generic language of message exchange that abstracts away from specific, lower-level implementations. In all these cases, either the implementation has no specific formal model, the model abstracts away from low-level implementations or the implementation of the EPP departs significantly from its formalisation (e.g., the model uses name synchronisation while the implementation uses more involved, lower-level technologies). Implementations of other frameworks based on sessions share similar issues (Hu, Yoshida & Honda, 2008; Hu et al., 2013; Neykova & Yoshida, 2014).

This is the first work that formalises how we can use choreographies in the setting of a practical communication mechanism used in SOC, i.e., message correlation. Our work gives the first correctness result for the compilation of choreographies to a language close to real-world implementations. More generally, our results are a reference for formalising the implementation of session-typed languages. In the future, this line of work may help to establish a certified choreography compilation. The principles behind the projection and neighbouring notions like realisability (which verifies whether, given a choreography specification, it is possible to build a distributed system that communicates exactly as the choreography specifies) and decomposition (which can enforce compliance by deconstructing a choreography into implemented and abstract behaviour, the former realised separately) of choreographies (Qiu et al., 2007; Carbone, Honda & Yoshida, 2012) sinks its roots in SOC/Web-services and research on ways to infer/realise/decompose interaction protocols such as message sequence charts (Alur, Etessami & Yannakakis, 2003, 2005), expanded in subsequent work (Busi et al., 2006; Montali et al., 2010; Basu, Bultan & Ouederni, 2012; Bravetti & Zavattaro, 2014; Basu & Bultan, 2016; Ancona et al., 2016; Hüttel et al., 2016; Scalas et al., 2017; Hennicker & Bidoit, 2018; Guanciale & Tuosto, 2019; ter Beek, Hennicker & Kleijn, 2020; Schewe, Ameur & Benyagoub, 2021; Coto, Guanciale & Tuosto, 2021; Barbanera et al., 2021; Cutner, Yoshida & Vassor, 2022; Vasconcelos et al., 2022; Dagnino, Giannini & Dezani-Ciancaglini, 2023; Castellani, Dezani-Ciancaglini & Giannini, 2023; ter Beek, Hennicker & Proença, 2023; Barbanera, Lanese & Tuosto, 2023). One concern of this strand of work is studying how the procedures for synthesis/verification of the implementations/choreographies characterise the category of the programs considered valid, e.g., what are the traits that discriminate projectable/realisable choreographies. For instance, as mentioned in “Properties”, of all the FC programs, we select only those that are well-typed (so that choreographies cannot end in deadlocks) and projectable (so that the projected components have enough information to faithfully implement the semantics of their source choreography).

Another distinctive trait of Applied Choreographies is a minimal realisation of asynchrony and out-of-order execution of independent actions via a swap relation, drawn from previous work on choreographic languages by Carbone & Montesi (2013) and Montesi & Yoshida (2013). Alternative approaches exist, e.g., Rensink & Wehrheim (2001) proposed a notion of partial termination which one can adapt (Edixhoven & Jongmans, 2022; Edixhoven et al., 2022, 2024) to reduce choreographies using a weak sequential composition, useful, e.g., to develop efficient implementations of the semantics of the choreographic language. Since we compile choreographic programs down to services, this aspect has a small impact on our work, but it can become relevant for future analyses on the semantics of choreographic programs. We believe that many models that use choreographies and sessions (or channel-based communications) can integrate our approach, including those based on process names (Carbone, Honda & Yoshida, 2012; Carbone & Montesi, 2013; Montesi & Yoshida, 2013; Honda, Yoshida & Carbone, 2016; Cruz-Filipe & Montesi, 2020; Cruz-Filipe et al., 2022, 2023; Cruz-Filipe, Montesi & Peressotti, 2023; Montesi, 2023) and on linear logic (Carbone et al., 2017; Carbone, Montesi & Schürmann, 2018). Our development shows that it is possible to keep a simple language model as a frontend, allowing developers to abstract away from how sessions are concretely implemented. Nevertheless, our Frontend Choreographies are expressive, as illustrated by our examples, and recent studies have shown that choreography languages such as ours are Turing complete (Cruz-Filipe & Montesi, 2020). Many works investigate how to introduce different features into choreographies, which we have not studied here and leave for future work. Examples include nested protocols (Demangeon & Honda, 2012), asynchronous two-way exchanges (Carbone, Montesi & Schürmann, 2018), and general recursion (Cruz-Filipe & Montesi, 2017) and the verification of properties on global recursive systems, e.g., that all sent messages can be received within a given bound (e.g., to avoid queue overflows) or that send actions within a given bound can execute (Heußner, Gall & Sutre, 2012; Basu & Bultan, 2016; Finkel & Lozes, 2017; Bouajjani et al., 2018; Lange & Yoshida, 2019; Bollig et al., 2021; Lagaillardie, Neykova & Yoshida, 2022). In our settings, both the capacity and number of queues are unbounded but, by using choreographies, we have pre-determined patterns of creation and usage, which future work on bounded queues can exploit to obtain efficient analysis routines.

The above features are orthogonal to our development, so their inclusion should be modular wrt our work. A feature found in other models that would require the extension of our contribution is the support for session delegation (Carbone & Montesi, 2013; Honda, Yoshida & Carbone, 2016). Delegation allows transferring the responsibility to continue a session from one process to another. Introducing delegation in FC is straightforward since we can just import the development from Carbone & Montesi (2013), Montesi & Yoshida (2013). Implementing it in BC and DCC would be more involved, but not difficult: delegating a role in a session translates to moving the content of a queue from one process (location) to another, and ensuring that future messages reach the latter. The mechanisms to achieve the latter part have been investigated in Hu, Yoshida & Honda (2008), which uses retransmission protocols. Formalising these “middleware” protocols and proving that they preserve the intended semantics of FC could be interesting future work. In the semantics of BC, we abstract away from how correlation keys are generated. This loose definition captures several implementations, provided they satisfy the requirement of the uniqueness of keys (wrt locations). Future work can implement languages, based on our framework, able to support custom procedures for the generation of correlation keys (e.g., from database queries, cookies, etc.). Another possible future direction is applying the results from this work to other models that support correlation and use alternative communication abstractions than channels, e.g., Linda-like tuple-based communications (Melgratti & Roldán, 2011; Pugliese & Tiezzi, 2012; Bruni et al., 2019; Basile et al., 2019). Generalising correlation, future directions can also include applications with attribute-based communication mechanisms (Alrahman, De Nicola & Loreti, 2019, 2020; De Nicola, Duong & Loreti, 2021).

Conclusion

In this article, we presented our framework of Applied Choreographies, which includes three calculi: a high-level choreographic language intended for developers, an intermediate-representation choreographic language, and a low-level, close-to-implementation distributed calculus. We equip our framework with a tight series of behavioural correspondences so that we guarantee that low-level distributed programs compiled from high-level sources faithfully follow their source specifications. By pairing our compilation with a type system and static checks that guarantee the absence of deadlocks in high-level choreographies, we obtain that the compiled distributed systems are deadlock-free. Specifically, we target service-oriented distributed systems that communicate over correlation mechanisms.

Besides the above contribution, Applied Choreographies introduce a novel semantics for choreographies that provides an abstraction for features of choreographies (message passing, creation of new sessions and processes) from their implementation (and the related complexity). To this end we (i) equip choreographies with a global deployment and (ii) define a separate semantics of effects on deployments. This separation allows us to compose our semantics of choreographies with other definitions of deployment and effects so that we have a straightforward way to capture different communication semantics (e.g., synchronous, asynchronous with buffers) and implementations (e.g., distributed objects as in Chin & Chanson (1991)). The notion of deployments allows us to formalise how choreographies can go wrong (see Section 1.3 of the Supplemental Material) and show that the theory of session types is useful not only to type communications on choreographies (Carbone & Montesi, 2013; Montesi & Yoshida, 2013) but also to check the correctness of deployments. Note that, except for the declaration of locations, Applied Choreographies has the same types and syntax from previous works Carbone & Montesi (2013), Montesi & Yoshida (2013), hence developers only have to specify protocols and choreographies and do not need to deal with deployment information or correlation data.

We have already mentioned some short-term future work in the previous section. More long-term projects include the investigation of compilation to other target languages/communication mechanisms based on correlation. For instance, those found in Erlang and Scala+Akka. Clearly, this would be a major development, since the actor-based concurrency and message passing of these languages are substantially different from that based on correlation, considered in this article. Another ambitious goal is the application of our research to the Internet of Things (IoT) setting. IoT promotes communication among heterogeneous entities—which use a wide range of communication media and data protocols—whose integration results in a cumbersome low-level programming activity. Indeed, to achieve a higher degree of interoperability, we propose the use of high-level, service-oriented languages for communication technology integration in IoT systems. In particular, an extension of Jolie by Gabbrielli et al. (2018, 2019) natively integrates the two most adopted protocols for IoT communication (CoAP and MQTT). Future steps on this approach can develop a variant of this work, specifically designed for IoT applications, that can then be compiled into the mentioned Jolie extension; allowing us to bring the correct-by-construction approach (through the formal correctness of compilation) developed in this work in the IoT field.

Supplemental Information

Supplemental Information 1 Supplementary Material.

Additional Information and Declarations

Competing Interests

Author Contributions

Data Availability

1 Operations are essential when communicating choices between the participants. A concrete example is a server offering a set of functionalities so that the client needs to annotate its message with the name of the functionality the message is intended for. In the example we present, we illustrate this situation when we use the operations ok and ko to signal on which conditional branch the interaction shall continue.

2 The (newque) and the from and to particles in (input), (choice) and (output) of Fig. 13.

3 Since the EPP acts on the syntax and FC and BC share the same syntax, distinguishing between them here is irrelevant.

4 Notice that ⊙ does not impose an ordering of the sequencing of actions B1, ... , Bn; this is fine for how we use ⊙ in Fig. 17, since we only need to impose an ordering among the whole blocks of actions ranged over by each ⊙.

The authors declare that they have no competing interests.

Saverio Giallorenzo conceived and designed the experiments, prepared figures and/or tables, authored or reviewed drafts of the article, and approved the final draft.

Fabrizio Montesi conceived and designed the experiments, authored or reviewed drafts of the article, and approved the final draft.

Maurizio Gabbrielli conceived and designed the experiments, authored or reviewed drafts of the article, and approved the final draft.

The following information was supplied regarding data availability:

There is no data or code for this publication.

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
