# Peer review of "A model for correlation-based choreographic programming"

_PeerJ Computer Science, doi:10.7717/peerj-cs.1907_

## Round 0.1 · original submission · Major Revisions

The reviewers agree that your paper provides interesting contributions. However, the first 2 reviewers have many suggestions on how to improve your paper, and how to improve the balance between contributions and the length of the paper. Note that they also comment on your references. Please make sure to also include relevant work of others. For a new version of this paper, please address these points.

·

Excellent Review

This review has been rated excellent by staff (in the top 15% of reviews)
EDITOR COMMENT
Thorough and detailed review, which will definitely help the authors to improve the review.

Basic reporting

The quality of the technical English is high, although some more technical presentation aspects could be improved.

The literature overview is very detailed, although it seems too focused on the author's own work and the literature of their close co-workers, since very few other (less than 20 years old) references are mentioned, even though more groups are currently working on choreographic languages and behavioural types.

The article structure and the figures all seem to have an overall high quality. However, I found the paper to be relatively long, and to lack both:
- enough examples throughout the paper, and
- a better introduction exemplifying the full picture, i.e. how a given program is manipulated until a DCC is produced.

The paper also includes enough detailed formal theorems and proofs, although not all very easy to follow (in my opinion).

Summarising, I'm happy with the quality of this paper, and I believe some parts can be improved. More details will be provided below.

Experimental design

The approach and methods used in this paper are theoretical and not experimental. The relevance of the research question is somehow subjective. Regarding relevance, the authors implicitly claim that the conceptual distance is smaller between
- the compiled calculus and some existing implementations of Service Oriented Architectures, than between
- existing choreographic languages for multiparty session types and their implementations.

The lack of an automatic (i.e., implemented) procedure between the input choreography and the final calculus, including type-checking (or type inference) also weakens this experimental aspect. Furthermore, it is not clear how computationally complex it can be to perform some operations that require exploring all congruent/equivalent terms.

In any case, my personal opinion is that this research question is of relevance for the scientific community (although maybe not yet for practitioners), and this work has been carried out rigorously.

Validity of the findings

The impact of this work is difficult to judge, due to its theoretical nature, and the novelty of each of the different steps of this approach is not overwhelming (in my opinion). However I find this work, as a whole, to provide a good and novel contribution, with well-formulated specifications and claims.

Additional comments

Summary
======
This work presents an approach to specify choreographies using a dedicated choreographic language (called Frontend Choreographies) that includes features such as sessions, service invocations (and instantiations), and data values sent through remote operation calls. These choreographies are then typed against a global session type, and compiled eventually into a calculus for service-oriented architectures (called Dynamic Correlation Calculus).

The authors use an intermediate semantics for choreographies in order to formally show that the strong properties provided by the session types are preserved until the compiled service oriented calculus.


General comments
================

This work is relatively long (i.e., it has many pages), and covers a relatively large variety of ideas. It is a good fit with this journal, and although it is purely theoretical (i.e., there is no implementation or automatization), it is (in my opinion) well aligned active research interests in the community.

As mentioned in the basic reporting above, I find this work of interest, although some aspects could be improved, mainly regarding the lack of guiding examples and some technical imprecisions.

More details, with concrete suggestions addressed to the authors, follow below.


Detailed comments to the authors
================================

** Better overview of the results with examples **

Partially due to the large size of this paper, a good motivation section will allow the reader to grasp the core of this paper without reading the full text. Currently the notion of a backend choreography and a DCC is only clear after reading the details throughout the paper.

The initial example in the introduction and the schema of the contributions in Fig. 1 help to provide some insights over the structure of this document, but not enough to guide through the many (sub)-languages used in this paper, and to grasp why they are different and needed.

My suggestion is to include a section (or a subsection of the introduction) to better capture this. More concretely:

1. to mention the different syntaxes and states used in the paper (see the table below as a suggestion),
2. to give a tiny example with a message sent and a service called (or more) written in each of these syntaxes (enough to highlight the differences), and
3. to explain what the user is expected to write initially (e.g., only the choreography, maybe also the global type, the locations of the services, etc.) and what he will get from this framework (e.g., the DCC expression, deadlock freedom guarantees.)

(Ideally also some possible disadvantages, not mentioned here, e.g., the fact that the typing system and "projectability" can restrict the expressivity of the choreographies, and potentially discard systems with a behaviour that can be perceived as correct.)

The possible table mentioned in (1) above could look like this:

```
Language: FC | BC | Type | DCC
Syntax: C | C | G | S
State: ⟨C,D⟩| ⟨C,D2⟩| G++ | S
```

Where
- `C` is the syntax for (frontend/backend) choreographies,
- `G` is the syntax for global types,
- `S` is a the syntax for a set of (active or not) parallel services,
- `G++` is an extended version with network state and locations of new processes,
- `D` captures the state of the network and the variable assignments of each process, and
- `D2` is based on `D` and captures the state of the network and processes with a stronger focus on locations, and using similar data structures to services `S` in DCC

Note that you present 4 operational semantics (one for each "language" mentioned in my table). In my view, an operational semantics can be given in terms of:
- what is a state
- what is the initial state
- what are the reduction rules
The table above only included the state, but you could easily hint at the other elements (although it may be too premature). Giving a more consistent presentation of these aspects for these 4 languages throughout the paper could improve readability.

Pushing this consistency idea even further, having the semantics of global types defined using a new structure (similar to `D`) to capture the network state, instead of your "runtime extension" (`G++`) would probably facilitate your formalisation and the flow of the paper. But I understand the preference to keep faithful to the literature, if that is the main reason.


** Projections **

Similarly to the idea of presenting the semantics in a consistent way, it may be possible to improve the presentation of projections of these languages (and maybe explain this in the beginning). E.g., you write:
- `[[G]]_A` to project global choreographies to roles, used to type choreographies
- `[[G++]]_A` and `[[G++]]_A^B` to extract network buffers from (extended) global choreographies
- `[[G]]_k` to project global projections to sessions (only used for session fidelity)
- `[[C]]_r` (and `C|_r`, which seems to be redundant) to project a choreography to the local choreography (which you call Endpoint Choreograph) of a process `r`.

More concretely, just as a suggestion, it could be good to mention upfront that these projections are defined, and what they are used for.


** "Projectable" choreographies **

You consider only well-typed choreographies in your encoding until DCC. A well-typed choreography C, although not explicitly defined, seems to be one for which a typing judgment exists that type-checks under its default "deployment" (initial state). Furthermore the typing rules calculate the projection of the global type to every role in the Start/Acc/Req rules.

Since these projections are partial (i.e., can fail), choreographies with no projections are not well-typed. Hence the concept of "projectable" in the end of the paper, in Section 6.4, seems redundant and unnecessary (unless I'm missing something).

Note also that this notion of projectability of choreographies, also known in the literature under other names (e.g., realisability, decomposability, implementability) is a relatively complex and studied topic. The approach taken by the Multiparty Session Types community that you use is relatively standard, and has the advantage of allowing a syntactic (and computationally simple) approach to obtain local implementations. However, this restricts the class of global behaviours that are valid (including projectable). This trade-off does not seem to be expressed in the paper; more generally, it would be good to understand earlier in the paper the class of "good" choreographies that you consider (including the ones that are well-typed using your MPST rules). More on this class of choreographies in the next topic.


** More restricted Choreographies? **

In order to produce a final DCC, it seems that you require the choreography to be a set of parallel "services", each starting either with "start" or with "accept". Furthermore, from the later encoding to DCC it seems that the parallel operator cannot occur in each of the processes, e.g., one cannot have the parallel inside an if-then-else.

When reading the paper, from the syntax and explanations, I was assuming that richer choreographies could be written, e.g., it would be possible to have the choreography C2 in Fig. 5 starting with a message received or a sent before waiting to accept a request. If this is not the case, then restricting the grammar (or clarifying upfront the class of choreographies of interest) would improve readability, and better prepare the reader for the target DCC language.


** Choice of syntax operators for choreographies **

- It is not clear why `k:c[C].product->s[S].buy(x);` instead of `k:c.product->s.buy(x);` (without the role), since it seems that these are introduced when a session starts. Is it possible to have the same processor name with different roles? If not, maybe the simpler notation is preferable.
- It is not clear why only the condition of an if-then-else does not have the role (e.g., `if b.confirm(...)` instead of `if b[B].confirm(...)`). Doest it mean that the same process name cannot be used over different sessions? (Note that here the session is not known, unlike with communication operators.)
- Similarly, many operators, such as local sends, use a different syntax over the different languages (`k:p[A].e->B.op(x)` vs. `&A.{op(U);...}` vs `op@lA(e) to lB`, and in the literature others also exist, e.g. `AB!op(e)`) - maybe using a more consistent variation of these syntaxes (e.g., a `->` notation in all languages) could help highlighting the conceptual differences of these languages (although I understand that being faithful to the literature is a valid reason to keep the notation different).


** Parallel operator **

Not all choreographic languages or behavioural types include the parallel operator (at least at the expression level), since it often makes analysis more difficult. E.g., the DCC syntax only supports the parallel operator at the level of services or processes, but not at the level of the behaviour of processes. Furthermore, global types have no parallel operator. However you decided to include the parallel operator on the behaviour level of your (frontend/backend) choreographies.

This mismatch is more explicit in the absence of the parallel operator in the compilation in Fig. 24, that should have been encoded (somehow). It is also visible when encoding in Fig. 11 the example of Fig. 4,5, where the system is a parallel composition of 2 choreographies (as mentioned in line 329), but the global type is a pair of types.

This redirects to my comment above on "more restricted choreographies", since maybe it would make sense to restrict more the usage of the parallel operator.


** Locations less clear **

While the location of processes is heavily used in DCC, it is less present in the choreography (and non-existent in global types). However, it would be good to have a better understanding of how they work via concrete examples from the beginning. It was not very clear to me how the typing of services `Γ,˜l: G⟨A|˜B|˜C⟩` was used, and even less the `p@l`. The latter seems to be missing in the typing rule, e.g., in the [T|start] rule, and I could not find where these are added or used (except maybe later in the encoding).
(I had to jump many times back and forth while reading the paper, and not all gaps became clear to me...)

For example, providing a complete `Γ` that could type a simple process with a service invocation, and use it to explain better how locations are managed, would help the reader to appreciate/understand how they relate to the locations in DCC.

My incomplete grasp of how locations are managed in typing context also limited my grasp of the (partial) coherence concepts.


** Weak sequencing **

You use a weak-sequencing operator `;`. This means that later interactions in a choreography can occur earlier, if they involve independent processes (as opposed to a strict sequencing operator). This makes sense since the target compilation uses a similar semantics. Technically you achieve this using a "swap relation", although other approaches exist in the literature, e.g., Rensink and Wehrheim notion of partial termination to reduce choreographies using a weak sequential composition ("Process algebra with action dependencies, 2001").

Your choice makes the operational semantics harder to automatise. I.e., discovering the next step can potentially involve traversing all possible permutations of independent actions. Probably this has a low impact on your work, since you only plan to animate the semantics of the projections (and not the global choreography), which use a strict sequential composition, and guarantee that their semantics coincide. A few words on this could be useful.


** Examples in Section 3.4 **

These examples are very welcomed here. However, many concepts and pages have passed since the previous example (Example 3, 9 pages before). Furthermore, these explain the previous definitions, while some examples can be more useful when preceding the formal explanations, motivating their need. Hence, having smaller simpler examples before this section would help the reader.

Furthermore, you have no example of a service call, and never describe a complete `Γ`. This could be also helpful. Note that it would be preferable, in Tables 1 and 2, to use the precise syntax for the `Γ`, e.g., writing `k[A]:end` instead of `k[A]=end`, and `˜l: A->B;.... <A|B,C|B,C>` (or a variation of this) instead of `G = A->B;...`.


** More examples! **

Later, when presenting backend choreographies, you have no example of a deployment (in the new deployment structure) nor the evolution of a deployment. Revisiting an existing example, now using this structure, would be very helpful.

This applies again for DCC. In fact, after Example 5 (page 21, §3.4) for FC and typing rules, the only examples that can be found are Example 6 (page 34, §6.1) for projecting choreographies, and Example 7 (page 37, §6.3) in the end of the paper for the encoded DCC service from Fig. 1. Exemplifying intermediate concepts, maybe before presenting these concepts if possible, would greatly improve readability.


** Data in BC (trees) **

In BC you explain how a key is used to add messages to a local queue and retrieve it later. At this point it is not clear why this is not a "session" key and instead a "correlation" key (maybe it is the same, just using a different name in a different context).

The big change is the usage of a tree structure for data, where nodes have no data, edges are labelled by a "label", and leafs are values or locations.

It is not clear why presenting this tree structure at this point. I guess the reason is because it is used by DCC, which is the final compilation target of this work, allowing this BC to be a middle point between FC and DCC. Otherwise, since this tree ends up being used only to represent state (in the "deployment"), this tree ends up being a relatively unstructured group of mappings (partial functions) or values. E.g.,
- `⟨Var⟩->Val` represent variable assignments in a process
- `⟨key,Role,Role,Var⟩->Val` represent variable assignments in a given channel in a process;
- `⟨⟩->Val` represent a value passed as argument of an operation (co-domain of `g_m` in the deployment);
- `⟨key,Role,"l"⟩->Loc`, where "l" seems to be a marker and not a specific location (which is confusing), to represent the location of a role in a session.

After some effort, we deduce that labels can be either keys, roles, variable names, or the marker "l".

Note that, in rule [D|Sup] of Fig. 18, the tree `t` is used but it is not defined. It might be obtained from `D(q_1)`, or might be a global tree assumed to exist (but this is missing). This also points to another data stored in the structure: `⟨Role,"l"⟩->Loc` to find the location of a role, and `⟨Role,Role⟩->Val` to retrieve the correlation key `t_ij` in a pair of roles (although this `t_ij` is also read from a tree, but I see no rule that adds this information.) These minor technical details should be fixed.

At this point, the name "x" and "y" are no longer variables but are *paths*. This adds some confusion - I would rather read "\pi" or similar, to avoid confusing variable names and paths.

Observe that trees are just a (somehow) more efficient data structure, that could be replaced here by something more structured (such as explicit relations or functions). Your choice for using less-structured trees is probably to align with DCC, and not to have a nice BC model (in my opinion). This reason (or whatever reason drove you to use trees) should have been mentioned here.

Another minor suggestion: I personally find easier to read `t(path)` instead of `path(t)` to retrieve a sub-tree, and `t[path->t2]` (or `t[t2/path]`) instead of `t◁(path,t2)`. (Feel free to ignore this suggestion.)

Yet another minor suggestion: instead of mapping locations to sets of processes in D, and "assume" that each location can run at most a process, it would be better to map each process to a location (or, if you prefer, map each process to a pair with a location and a tree, combining `g_l` and `g_s`).


** Smaller issues **

- The global type of the example from Fig. 2 does not seem to capture the call to the service. This raises the question of why it is enough to check the type against this choreography, since there seems to be no relation between interactions of the server in different sessions (and it is not clear to me why this is not a problem).

- Since you have recursion, a problem studied in the literature is related to how to guarantee that an infinite amount of sends (and maybe service invocations in your case) cannot be performed without being received. This is important to avoid, e.g., queues to overflow. In many cases it is possible to provide sufficient conditions that guarantee this property. A few words on this property would be appreciated.

- The meaning of "correlation" could have been clearer from the beginning. I have learned the concept of correlation in statistics, but it is not made clear what "correlation mechanisms based on message data" is. Since the term appears in the title, and later in "correlation keys", it would be good to clarify what it means early in the paper.

- The term "deployment" does not seem to be the correct one here (at least for me). Software deployment has to do with the process of releasing, installing, testing, executing, etc. It seems to me that a term like "configuration" (popular in term-rewriting community) or "state" (popular when describing operational semantics using transition systems) would be more appropriate. In any case, I leave this choice to the authors, although a brief explanation of why the term would be appreciated. Similarly, "default deployment" seems that should be the initial configuration/state. Similarly (again), the term "runtime typing" reminds me "runtime types" (inferring types at runtime), instead of typing configurations/states, but again I leave this naming choice to the authors.

- Why not communication (`k:a[A]."ok"->b[B].fwd(x);C`) as syntactic sugar (for `k:a[A]."ok"->B.fwd; k:A->b[B].{fwd(x);C}`)? Having syntactic sugar such as this would make your rule-set smaller, without compromising expressivity in this case.

- In Fig. 24 you use a generalised form of sequencing `⊙`. These generalisations typically require the base operator (in this case `;`) to be commutative (and associative and with an identity). However sequencing is not commutative, unless the elements are independent (since you are using weak-sequencing). It is worth mentioning why the order is not important (if that is indeed the case).


** Minor issues **

- line 101, you mention a core question of whether we can use choreographies to prove that a concurrent, distributed program will execute only its intended sequence of interactions. I think you mean "*all and only* its intended sequence of interactions."

- Fig. 14, the first 3 typing rules are very similar, and this could be highlighted by writing them using a more similar structure. E.g.:
+ writing either `˜l: G⟨_|_|_⟩ ∈ Γ` above or `Γ,˜l: G⟨_|_|_⟩` below,
+ avoiding defining `˜r[C]`, and replace it by the definition `p[A],˜q[B]`,
+ writing `p : k[A],k[A] : [[G]]_A` using `init` in Req.

- Table 1, there is a typo in `C'` definition: `first` and `second` should be `pass` and `fwd`.

- Def. 5, you meant to join the 2 sets with union (U) instead of a comma (,).

- Line 759, when referring to the rules for updating a deployment in Fig. 18, would be useful to recall that the semantics of BC is given by Fig. 7, as in FC.

- Def. 7, you use an algorithm to compute `⟨⟨D⟩⟩^Γ`. I would expect a more traditional (mathematical) definition, e.g., `⟨⟨D⟩⟩^Γ = {l -> P | l@p ∈ Γ, P={p|l'@p∈Γ}} U {...}`. Algorithms usually make sense to demonstrate the computational tractability of a formula, or the efficiency of an approach, but they target more the implementation than the concept. Maybe you could justify why using an algorithm here (maybe for compactness), or rephrase it as mentioned above.

- Section 6.4 - it would be useful to recall explicitly where the operators used in Theorem 6 are defined (maybe in the paragraph preceding Theorem 6).

- Overall, you could have been consistent when inlining enumerations (e.g., always a)b) or i)ii))

- Minor suggestion: maybe writing in LaTeX `\ell` instead of `l` for locations would look more elegant and readable.

Cite this review as

Reviewer 2 ·

Basic reporting

SUMMARY:

The authors present the formal framework of Applied Choreographies, which they originally introduced in their FORTE 2018 conference paper, and which encompasses three stages of compilation involving the following three calculi:
- the (high-level) calculus of Frontend Choreographies (FC) presented in Section 2
- the (intermediate) calculus of Backend Choreographies (BC) presented in Section 4
- the (low-level) calculus of Dynamic Correlation Calculus (DCC) presented in Section 5
The main contribution claimed by the authors is "a behaviour-preserving compiler from Frontend Choreographies to DCC distributed services/processes", presented in Section 6. The compiler is paired with a type system, presented in Section 3, and static checks that guarantee the absence of deadlocks in high-level choreographies, and, as a consequence, also in the compiled distributed systems. The presented results are technically executed by the authors by a composition of two translations, one from FC to BC and one from BC to DCC via a transformation relying on a typical projection operation that maps FC to (local) endpoints. BC is basically a different interpretation of the syntax of FC: instead of channel-based communications it adopts correlation-based communications. In fact, the authors notice that one major contribution is the correlation-based semantics, which they claim to be "close to real-world implementations" - closer than channel-based communication. The authors extended their conference paper by providing full formal definitions of the calculi, more detailed examples, and (many, but not all!) full proofs - more below.


JUDGEMENT:

The contribution presented in this paper does not seem to provide any new general results. The contents of Sections 2 and 3 are (as also noted by the authors) a quite straightforward variation of the content of Montesi and Yoshida (2013) and the delta seems to be rather minimal. So, the core result is to be found in the transformation from BC to DCC. However, the mapping from channel-based to correlation-based communication is not difficult, and it seems to follow existing approaches - more below. In particular, Theorem 4, which shows the correspondence between FC and BC, becomes fundamental, but strangely enough its proof is only sketched. It is the only theorem with merely a proof sketch. There are a few lemmata with proofs that are only sketched, but since the provision of "full proofs" is claimed as one of the extensions of this paper, with respect to its conference version, this is really disappointing and a missed opportunity. To make things worse, Theorem 4 is heavily used in the proof of Theorem 6. The authors should make the proof of Theorem 4 solid, also because the proof of Theorem 6 is quite demanding as it is (interleaving it with definitions and additional lemmata is not helping).

In my opinion, the paper does not provide sufficient new contributions to warrant its current size (93 pages!). The current presentation is a mixture of many different types of contributions. The first transformation appears to be a careful adaptation of previous work, and the typing infrastructure it enables mirrors the one in previous work. The authors could have focused on the second transformation plus Theorem 4, fully proved, which does not require the typing. Section 3 on typing could either be moved to the Appendix or be made available as auxiliary material on some public repository. I fail to see its need for showing the correctness of the compilation per se (a choreography that deadlocks would simply be compiled into a DCC program that deadlocks). This would considerably improve the reading of the paper. Another problem that I have with this paper in its current state is that I find it too much focussed on the body of work by the authors (and friends). I miss relevant related work - more below. And I would have liked to read about how their approach could be used in other frameworks for choreographies, like the ones mentioned below. The paper is very self-centered: almost 50% of the references to other scientific publications are self-references, and Montesi and Carbone (2011) is referred to as "a standard formal model for Service-Oriented Computing" (a bit of an overstatement, in my opinion).

Experimental design

The paper fits in the scope of PeerJ and the research question "Can [we] define a formal model of choreographies based on message correlation?" such that "the complexity of implementing communications should not leak into the choreographic programming model exposed to programmers, and should just be a "detail" that we can forget about with confidence", is worthy of investigation.

Validity of the findings

The paper is very well written and also the presentation is nice and well worked out. The Appendix is less accessible and, as mentioned above, some of the proofs were quite a struggle, also due to their interleaving with auxiliary definitions and lemmata - and I am also not entirely sure I managed to grasp all of them in full detail. I am not an expert in process calculi.

Additional comments

GENERAL COMMENTS:

In ll. 69, 122, and 416, you refer to Multiparty Session Types, but each time with different references - please be consistent. Moreover, in l. 512, suddenly the capitalisation is removed: multiparty session types - again, please be consistent.

On p. 5, when you describe Fig. 1, I suggest to invert the paragraphs "EPP ... Choreographies" and "Backend ... Choreographies".

In Sect. 7, there is quite some apparently related work missing:
Other process-algebraic frameworks like Klaim and CoWS, which are not based on channels but feature Linda-like tuple-based communications, which have moreover been equipped with correlation data (e.g., https://doi.org/10.1007/978-3-030-21485-2_11, https://doi.org/10.1016/j.jal.2011.11.002, https://doi.org/10.1007/978-3-030-30985-5_9, https://doi.org/10.1007/978-3-642-30065-3_13).
Furthermore, attribute-based communication seems to generalise correlation-based communication (cf., e.g., https://doi.org/10.1016/j.scico.2020.102428, https://doi.org/10.1016/j.ic.2019.104457).
I also found an apparently compilation-preserving framework for choreographies (viz., https://doi.org/10.1016/j.scico.2020.102498) and a provably correct implementation of non channel-based calculi (viz., https://doi.org/10.1016/j.scico.2020.102567).

The References are sloppy:
javascript object notation (json) -> JavaScript Object Notation (JSON)
Extensible markup language (xml) -> Extensible Markup Language (XML)
References "Carbone, M., Montesi, F., and Schürmann, C. (2017a)" and "Carbone, M., Montesi, F., and Schürmann, C. (2018)" are the same.
Reference "Cruz-Filipe, L. and Montesi, F. (2020)" is the extended journal version of "Cruz-Filipe, L. and Montesi, F. (2016)." - it suffices to cite the former.
Lafuente, A. L. -> Lluch Lafuente, A.
Reference "Gabbrielli, M., Giallorenzo, S., Lanese, I., and Zingaro, S. P. (2018)" is said "to appear", but I presume it has appeared by now...
iot -> IoT (2x)
Reference "Honda, K., Yoshida, N., and Carbone, M. (2016)" is the extended journal version of "Honda, K., Yoshida, N., and Carbone, M. (2008)" - again, it suffices to cite the former.
java -> Java
Soap -> SOAP
facebook -> Facebook
Unified modelling language -> Unified Modelling Language


DETAILED COMMENTS:

l.51: behaviours -> behaviour (uncountable noun)
l.79: respectively -> , respectively,
l.83: Concluding -> Returning to
l.155: guarantee -> guarantees
l.156: Can define -> Can we define
l.180: (Section 2) programs -> programs (Section 2)
l.185: behaviours -> behaviour
l.211: Service-Oriented Computing -> SOC
l.222: works -> work
l.246: service-oriented computing -> SOC
l.247: in here -> in this article
l.420: In Section 3.1 we -> In Section 3.1, we
l.423: In Section 3.2 we -> In Section 3.2, we
l.428: In Section 3.5 we -> In Section 3.5, we
l.454: checking -> Checking
l.480: ; -> ; or
l.531: works -> work
l.540: behaviours -> behaviour
l.573: what presented -> what we presented
l.576/577: Type checking -> Type Checking (paragraph header)
l.584: are evolution of -> are evolutions of
l.610: report in -> report the following in
l.616: in D -> in deployment D
l.633: reduction -> reductions
l.636: raw -> row
l.654: appendix -> Appendix
l.655: behaviours -> behaviour
l.656: We denote -> We denote by
l.666: appendix -> Appendix
l.675: appendix -> Appendix
ll.679-680: Service-Oriented Computing (SOC) -> SOC
l.681: semantics rules -> semantic rules
l.692: unbound number -> unbounded number
l.715: addressee service -> addressee's service
l.717: state -> State
l.723: represents state -> represent states
l.735: in the reminder -> in the remainder
l.806: same syntax of -> same syntax as
l.833: point -> point to
l.848: defied by -> defined by
l.858: also for -> also to
l.876: change -> changes
l.894: behaviours -> behaviour (3x)
l.895: behaviours -> behaviour
l.895: define -> defines
Fig.20: Behaviours -> Behaviour
l.896: behaviours -> behaviour
l.896: use -> uses
l.903: Behaviours -> Behaviour
l.939: same of -> same as
l.940: point an -> point to an
l.941: pointed by -> pointed to by
l.980: on all -> onto all
l.984: written -> denoted
l.985: results into -> results in
l.991: Let us -> We
l.997: written [[C]]_p -> remove (cf. l.984)
l.1002: projected into -> projected onto
l.1009: X, indeed -> X; indeed
l.1012: behaviours -> behaviour
l.1013: in Appendix -> in the Appendix
l.1021: yeld -> yield
l.1027: the paragraph -> this paragraph
l.1032: in Appendix -> In the Appendix
l.1046: behaviours -> behaviour
l.1050: introduce -> we introduce
l.1061: appendix -> Appendix
l.1065: as here -> as in this article
l.1068: behaviours -> behaviour
l.1075: appendix -> Appendices
l.1075: briefly -> In short
l.1080: w.r.t. -> wrt
l.1087: follow -> follows
l.1101: environment \Gamma -> environment \Gamma, then
l.1106: behaviours -> behaviour
l.1148: appendix -> Appendix
l.1159: Service-Oriented Computing (SOC) -> SOC
l.1184: features to -> features into
l.1194: which use -> which uses
ll.1198-1199: wrt to -> wrt
l.1208: Service-Oriented -> service-oriented
l.1210: contribution above -> above contribution
l.1216: objects Chin -> objects as in Chin
l.1217: let us -> allow us to
l.1222: have only to -> only have to
l.1226: besides correlation-based ones-orientation -> rephrase
l.1228: paper -> article
l.1231: low level -> low-level (cf. high-level in l.1232)
l.1234: Gabbrielli et al. (2018, 2019) -> remove (cf. l.1232)
l.1379: appendix -> Appendix
l.1382: appendix -> Appendix
l.1384: appendix -> Appendix
l.1399: appendix -> Appendix
l.1455: \qed should be right-justified
ll.1455-1456: Below we restate ... in the proofs -> please provide empty lines before and after
l.1461: stronger result Theorem 7 -> stronger result of Theorem 7
l.1462: define -> state
l.1632: We report below ... in the proof -> please provide empty lines before and after
l.1732: behaviours -> behaviour
l.1747: behaviours -> behaviour
l.1765: property -> Property
l.1919/1920: appendix -> Appendix
l.2155: We report below ... of D,C. -> please provide empty lines before and after
l.2197: Definition 12 l, t_c, and t_m -> Definition 12, l, t_c, and t_m
l.2222: Definition 12 t_c and t_m -> Definition 12, t_c and t_m
l.2242: From Definition 12 we have, let ... -> please rephrase
l.2253: From Definition 12 we have, let ... -> please rephrase
l.2267: comment the -> comment on the
l.2291: behaviours -> behaviour
l.2323: From Definition 12 we have, let ... -> please rephrase
l.2395: From Definition 12 and Theorem 4 we have, let ... -> please rephrase
l.2476: where, let -> please rephrase
l.2480: such that, let -> please rephrase
l.2483/2484: where, let -> please rephrase
l.2513: M(t_c) = m -> M(t_c) = m, then
l.2572: the sentence extends too much into the right margin

Cite this review as

Reviewer 3 ·

Basic reporting

+ Paper is well-written and clear. The authors made use of many examples which was appreciated.
+ Background is thorough and does a good job contextualizing the work within the recent literature
+ Figures and code snippets are excellent and well-explained.
+ The paper is self-contained, it makes references to the authors' prior work, but it summarizes the results of that work and how this paper builds on them.
+ Theorems are defined and stated. Using a figure to explain how each theorem fits into the overall project was useful (Figure 1). Proof sketches are provided, with full proofs in the appendix.

Experimental design

+ Research is in scope for the journal.
+ Research question (how to take a simple choreography language and compile it to a service oriented framework with formal guarantees) is interesting and novel. The introduction and background section address how this paper goes above and beyond prior work (including a conference version of the paper) and fills a gap in the literature (the gap is the lack of correctness results for choreographic programming).
+ The paper is thorough - every phase in the translation of a choreography to an intermediate representation to the target language is rigorously defined and explained. The correctness of every phase of the translation is defined and proved.
+ This is theoretical work and full proofs are provided, so no concerns about replication.

Validity of the findings

+ The authors clearly state how others could build on their work to prove other interesting results (session typed computation, internet of things, etc). They also discuss how they could build on their current work.
+ All proofs are provided.
+ Conclusion effectively summarizes the paper and does not overstate the results.

Additional comments

This was an excellent paper that I enjoyed reading.

Minor comments:

21: "reason about"
101: "centers around"
129 "exactly the communications"
135 "The unifictation of the two approaches was made possible by ..."
165: "focusing"
417: "same syntax as"
418: inconsistencies with states?
531: "with respect to prior work"
538: "with respect to a choreography if for example"
540: "such as deadlocks"
575: "we do not swap a send operation that is causally dependent on a send operation on the same role"
576: "Runtime Type Checking and Typing Rules"
576: "The extension consists of ..."
582: "associated with one location"
597: "associates ...with.."
598: "with respect to"
600: "We state the formal definition below."
678: "similar" instead of close
678: "communicate"
687: "in the following section"
712: "abstract away"
735: "in the remainder of the paper"
737: "a Backend or Frontend choreography"
760: "We explain the rules below."
808: "With respect to"
820: "Then, we make the following updates:"
871: "which informs how we can apply..."
975: "Fortunately, CC and DCC are similar enough that supporting..."
977: "We also recall that.."
981: "an adaptation of the one presented..."
1001: "if participated in"
1034: "Definition 10 states that the EPP of a choreography is...."
1049: With respect to
1172: "More generally our results are a useful reference for formalizing the implementation of session typed languages"
1181: "abstract away from"
1183: "such as ours"
1197: "abstract away from"
1211: "provides an abstraction for"
1228: "languages based on correlation"
1231: "whose integration results in"
1237: "import to"

Cite this review as

---

## Round 0.2 · Minor Revisions

Per previous discussion with Jackie Thai, please remove the Appendix and resubmit so this submission can be re-evaluated for length.

·

Basic reporting

This is a revised paper that I had previously reviewed. The previous version of this manuscript had already, in my view, a very high quality. I believe that this revised version improves the previous one and that all my main concerns were successfully addressed. More specifically, I appreciated:

- the move to the appendix some aspects that were breaking the flow of the story, namely of the typing rules; and
- the inclusion at an early stage of how a simple example is encoded at different phases of the proposed encoding.

Furthermore, the explanations in response letter were very detailed and helped clarifying several of my own doubts.

I include below a few more details regarding the early example, but these should be regarded only as suggestions and not required changes.

Experimental design

I have nothing to add to my previous review.

Validity of the findings

I have nothing to add to my previous review.

Additional comments

General comments
================

I am very pleased with the revised version and with the effort of the authors to address and respond to all the reviewers' comments, including mine.

As mentioned in the basic reporting, I find the main changes to this manuscript to be a welcomed improvement. I therefore recommend acceptance of this paper. I include below only a few personal comments regarding mainly the initial example, which are only suggestions and not required changes.


Detailed comments to the authors
================================

I find the new Table 1 to be very useful. I include only a few comments and personal suggestions regarding it.

- You use a single header (FC, BC, DCC) and a separation halfway ("process c sends its message"). I would suggest to name both parts, and clarify better the 2nd part. E.g.:
+ 1st part - Initial program (or maybe after the "start")
+ 2nd part - After "c" sends a "product" message

- These names for each part can also be written on the left- or right-side, similar to "deployment" and "program"

- It seems that, from FC to BC, the location names were introduced. This is because they were initially in the program (in the start operator), but disappeared in the FC case. Maybe emphasising that your top program is not the initial state, but after the session agreement (with a "part-name" that reflects this).

- The program on top and below are, in fact, different "states" of the same program (or one is the evolution of the first). But at first sight it seems that, instead, the state of the program is captured in the deployment section (based on the code comments). Maybe writing "state of deployment of c" instead of "state of process c" could help.

- Suggestion: in DCC, after sending: add a comment stating that "Bc, Tc, BS, and Ts remain unchanged from above"; also swap the order of the service network and Qs.

- Suggestion: in the deployment after sending, for the FC and BC, also add something like "unchanged from before".


The appendices are very extensive (even more now), and I did not go through the details of the proofs. But I realised that some blocks were unnecessarily small (some of which have been fixed in this version). Namely in page 57/104 and 59/104. Improving readability of these blocks could be helpful.


== Minor issues ==

- You write MST as a shorthand to Multiparty Session Types. I find both MST and MPST in the literature, although I have the impression that MPST is more common (just a personal opinion).

- In section 2.2.1 you use the name "Frontend deployment" (small case "d") in most places (the title uses capital "D" following your title-convention). I'm not sure the small case "d" is on purpose.

Cite this review as

Reviewer 2 ·

Basic reporting

I am satisfied with the meticulous revision that the authors have performed. In particular, the size of the paper without appendix is more in proportion to the contribution, and its readability has much improved (also due to the example in the Introduction, added in answer to comments from one of the other reviewers).

Moreover, Theorem 4 (now Theorem 1) is more solid, implying also Theorem 6 (now Theorem 3) is more convincing (though still a little hard to follow, due to its interleaved presentation).

Finally, the related work section has been expanded with relevant related work. However, triggered by a comment from one of the other reviewers, concerning the relation between projectability of choreographies as studied in this paper, and neighbouring notions known in the literature as decomposability, realisability, or implementability, it would be worth citing recent papers by Hennicker et al. ( https://doi.org/10.23638/LMCS-14(1:1)2018, https://doi.org/10.1007/978-3-030-64276-1_11, and https://doi.org/10.1007/978-3-031-47963-2_15) which seem to be highly relevant.

Experimental design

See basic reporting.

Validity of the findings

See basic reporting.

Cite this review as

---

## Round 0.3 · accepted · Accept

The appendix has been moved into the supplementary material and all the remaining comments have been addressed.